# Heterogeneous Graph Temporal Fusion Transformer for Time Series Forecasting in Multi-Domain Physical Systems

## Abstract

Existing Transformer-based models effectively capture multivariate dependencies, while pre-trained large models achieve strong generalization but are often confined to single-object or single-physics settings. Spatial-temporal approaches leverage graph structures but fall short in modeling heterogeneous entities with diverse inter-variable interactions, and they often lack mechanisms to enforce physical consistency. To address these challenges, we propose the Heterogeneous Graph Temporal Fusion Transformer (HGTFT), a pre-training and fine-tuning framework tailored for spatially and temporally structured physical environments. HGTFT tokenizes observation points and generates embeddings that capture both temporal patterns and spatial correlations, enabling the integration of heterogeneous static and dynamic information. We further introduce optimized normalization and physics-informed loss functions that enhance predictive accuracy while improving physical plausibility. Applied to temperature, flow, and energy-related datasets in building environments, our approach demonstrates strong zero-shot generalization and achieves substantial accuracy gains through few-shot fine-tuning with domain-specific data.

## 1 Introduction

Energy is a key factor in the development of AI, while energy systems are typical multiphysics systems involving coupled thermal, fluid, and electrical processes. More generally, multi-domain physical systems such as power grids and building energy networks consist of heterogeneous entities (e.g., sensors, actuators, and control devices) that interact across multiple physical fields. Accurate forecasting in these systems is critical for efficiency, safety, and sustainability, yet remains challenging due to diverse data modalities, complex structural dependencies, and domain-specific physical dynamics. Existing approaches fall into four categories.

**Physics-prioritized/analytical methods:** These use known governing equations and boundary conditions directly. They provide strong interpretability but often struggle with complex domain coupling or computational cost in high-dimensional/irregular settings.

**Physics-guided neural solvers:** Physics-Informed Neural Networks (PINNs), e.g., DeepXDE Lu et al. (2021) and Neuromancer Drgona et al. (2023), embed PDE residuals, boundary conditions, or conservation laws into neural networks for strong physical consistency. Yet they incur high derivative costs and are limited to continuous field simulations, not heterogeneous multi-entity systems.

**Purely data-driven methods:** Time-series models, e.g., LSTM and LTMs (Dong et al. (2024), Liu et al. (2024c)), capture general temporal patterns but lack physical grounding, making extrapolation or safety-critical predictions unreliable. LLM-based time-series methods (Zhou et al. (2023), Liu et al. (2024b)) inherit this limitation, as tokenization treats inputs as numerical patches or sequences, ignoring spatial structure, static context, and multi-entity physical dependencies.

**Data-driven methods with physical constraints:** These approaches are primarily data-driven, leveraging network architectures to encode heterogeneous information while incorporating physics through appropriate loss functions and training pipelines. By doing so, physical consistency can be learned and enforced within the model. We propose the Heterogeneous Graph Temporal Fu-

sion Transformer (HGTFT), a framework that integrates heterogeneous spatial-temporal information while maintaining consistency with underlying physical constraints. In HGTFT, each token represents a node's state at a time step, embedding both dynamic and static attributes. The encoder captures temporal dynamics and heterogeneous spatial relations, while the decoder adapts these representations to task-specific forecasting. A tailored training pipeline incorporating domain-informed loss functions and improved normalization enhances predictive accuracy without compromising physical validity. Our contributions are threefold: (1) We define heterogeneous graph forecasting in multi-domain physical systems, covering a wide range of real-world infrastructures. (2) We introduce HGTFT with tailored tokenization and embedding strategies that fully leverage graph structures, static attributes, and dynamic features. (3) We develop a physics-informed training pipeline with improved normalization, enabling HGTFT to achieve higher accuracy and stronger consistency with domain physics across benchmarks.

## 2 RELATED WORK

**Transformer for Time Series Forecasting.** The Transformer model Vaswani et al. (2017) has revolutionized time series forecasting via attention. Extensions like Informer Zhou et al. (2021) use probabilistic sparse attention, while Frozen Pretrained Transformer Zhou et al. (2023) adapts pretrained models from other domains, linking self-attention with principal component analysis. For multivariate forecasting, Crossformer Zhang & Yan (2023) implements a two-stage attention mechanism for temporal and cross-dimensional dependencies, and Temporal Fusion Transformer (TFT) Lim et al. (2021) provides interpretable mixed-input forecasting. Recent approaches like Time-Siam Dong et al. (2024) and Timer Liu et al. (2024c) use unlabeled data for representation learning. Unified training paradigms Woo et al. (2024) allow single models to handle multiple tasks, while decoder-only models Das et al. (2023) enhance prediction efficiency. These efforts demonstrate the potential of pre-training to improve generalization and accuracy in time series forecasting.

**Spatial-Temporal Forecasting.** Graph Neural Networks (GNNs) have made strides in graph-based learning through structural and positional encoding. Approaches like LSPE Ying et al. (2021) and NodeFormer Wu et al. (2022) address scalability, while LETR Xu et al. (2021) and Molecule Attention Transformer Maziarka et al. (2020) apply Transformers to specialized tasks. For heterogeneous graphs, HDGT Jia et al. (2023) and HAN Wang et al. (2019) use hierarchical attention to capture diverse node and edge types. In spatiotemporal forecasting, DCRNN Li et al. (2017) models spatial diffusion and temporal dependencies using diffusion convolution within a recurrent framework, while STEP Shao et al. (2022b) extends this approach with a pre-training enhanced GNN for long-range temporal patterns. Spacetimeformer Grigsby et al. (2021) and heuristic graphs Shao et al. (2022a) model complex temporal-spatial sequences, while Graph Neural ODEs Poli et al. (2019) incorporate differential equations for capturing dynamic temporal dependencies. Models like STS-GCN Song et al. (2020) and STSGT Banerjee et al. (2022) combine GCNs and Transformers to model synchronous spatial-temporal dependencies, applied to traffic and pandemic forecasting. Architectures such as HGT Hu et al. (2020) and PromptST Zhang et al. (2023) leverage pretraining and adaptive tuning for heterogeneous, multi-attribute graph predictions. Similarly, UniST Yuan et al. (2024) employs prompt-based learning and extensive pre-training to enhance generalization in urban spatio-temporal prediction.

**Large Language Models for Time Series.** Large Language Models (LLMs) have been adapted for time series tasks, particularly in few-shot and zero-shot settings. TimeGPT-1 Garza et al. (2023) reprograms LLMs for time series prediction by aligning embeddings with time-domain features, while Gruver Gruver et al. (2024) demonstrates zero-shot forecasting without fine-tuning. LLM4TS Chang et al. (2024) and TIME-LLM Jin et al. (2023) optimize LLMs for time series, improving adaptability to specialized datasets and temporal patterns. TimeCMA Liu et al. (2024a) introduces cross-modality alignment to enhance temporal understanding, and TimeChat Ren et al. (2024) expands this to multimodal contexts, integrating temporal information for applications like video understanding. These studies highlight LLMs' potential as general-purpose forecasters, though challenges in temporal representation, data efficiency, and interpretability remain.

**Physics-informed methods.** Another research line integrates physical constraints into neural networks to enhance interpretability and consistency with known dynamics. Representative frameworks such as PINNs Raissi et al. (2019), DeepXDE Lu et al. (2021), and Variational PINNs Kharazmi

et al. (2019) enforce the governing differential equations via loss regularization. Extensions including Neural Operators (Li et al. (2020b); Li et al. (2024)) and Fourier Neural Operators (Li et al. (2020a)) learn mappings between function spaces for efficient PDE simulation, while hybrid methods like Graph-based PINNs Gao et al. (2022) aims to connect continuous physics with graph structures.

## 3 PROBLEM DEFINITION

**General Spatial-Temporal Forecasting Problem.** Spatial-temporal forecasting problems, such as those involving traffic networks, the COVID-19 pandemic, or power grids Guo et al. (2019); Banerjee et al. (2022); Liu et al. (2023b), can typically be formulated using a spatial network $G = (V, E, A)$, where $V$ is the set of node vertices, $E$ represents the set of edges, and $A$ is the adjacency matrix describing relationships between nodes. The goal is often to predict future observations for a single node type with a single relationship type. Each node entity $v_i$ in the graph is associated with a graph signal matrix $X(t)_G \in \mathbb{R}^{N \times F}$, where $F$ is the number of features per node, and $t$ denotes the time step. $X(t)_G$ captures the spatial network observations at time $t$, with each entry $X_{i,t}$ representing the feature vector of node $v_i$ at time $t$. The task is to predict future spatial-temporal data by learning a mapping function $\mathcal{F}$ that maps historical series $\{X(t - T_{\text{past}} + 1)_G, \ldots, X(t)_G\}$ to future observations $\{X(t + 1)_G, \ldots, X(t + T_{\text{future}})_G\}$, where $T_{\text{past}}$ is the length of historical data and $T_{\text{future}}$ is the forecast horizon.

**Extension to Heterogeneous Graph Forecasting in Multi-Domain Physical Systems.** In contrast, our problem involves a more complex heterogeneous graph comprising multiple node types and relationships. Each node $v_i$ is characterized by static attributes $s_i$ and time-variant features, which are further grouped into: 1) variables known for both past and future $x_i$, 2) variables known only for the past $z_i$, and 3) the prediction variable $y_i$. The problem can be formulated as:

$$\hat{y}_{i,t+1:t+T_{\text{future}}} = \mathcal{F}(s_i, x_{i,t-T_{\text{past}}+1:t+T_{\text{future}}}, z_{i,t-T_{\text{past}}+1:t}, y_{i,t-T_{\text{past}}+1:t}, N(v_i)), \quad (1)$$

$$N(v_i) = \bigcup_{r_l \in R} N_l(v_i), \quad (2)$$

where $\hat{y}_{i,t+1:t+T_{\text{future}}}$ denotes the predicted target sequence for node $v_i$ over the future horizon $[t + 1, \ldots, t + T_{\text{future}}]$, and $\mathcal{F}$ is the learned forecasting function. $N(v_i)$ aggregates neighborhood information for node $v_i$ across relation types $r_l$.

This extension is significant due to its ability to model complex multiphysics systems with diverse node types, features, and interrelationships, prevalent in real-world applications such as nuclear reactors, aerospace vehicles, biomedical devices, combined heat and power systems, and smart buildings. These systems necessitate advanced forecasting models capable of capturing intricate interdependencies and dynamic interactions across different physical domains. For illustrative examples and a discussion on the necessity of sophisticated modeling approaches, refer to Appendix A.

## 4 MODEL ARCHITECTURE

The proposed HGTFT model, outlined in Figure 1, is designed for the previously defined problem by aggregating multi-dimensional data across static and dynamic node features within a heterogeneous graph structure. Features are aligned into unified embeddings per entity and time point, and these embeddings pass through neural layers that aggregate information across temporal and graph dimensions, resulting in fixed-dimension representations. The representations are then forwarded to task-specific model layer for dimension transformation tailored to each task.

**Fusion Layer.** Each node $v_i$ is associated with static covariates $s_i$ and time-varying features: future-known variables $x_{i,t}$, past-only variables $z_{i,t}$, and target variable $y_{i,t}$. We first map all available inputs into a shared $d$-dimensional latent space and fuse them using a Variable Selection Network (VSN) inspired by TFT Lim et al. (2021). This produces a time-dependent node representation:

$$h_{i,t}^{\text{node}} = \text{VSN}(\text{Proj}(s_i), \text{Proj}(x_{i,t}), \text{Proj}(z_{i,t}), \text{Proj}(y_{i,t})), \quad (3)$$

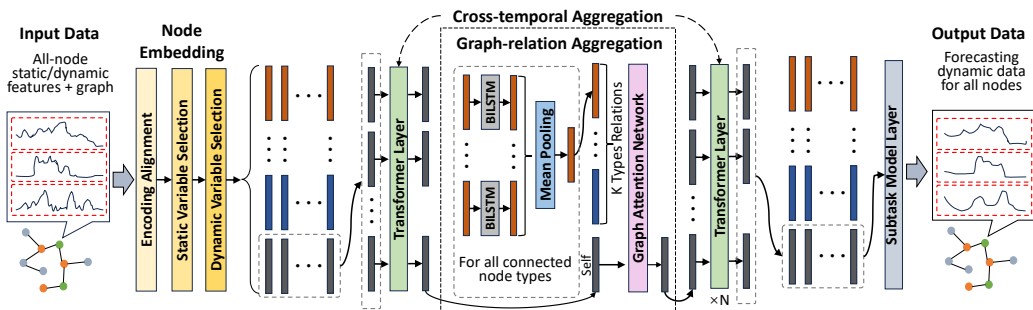

Figure 1: Architecture of the proposed HGTFT. The Fusion Layer converts heterogeneous data into unified-dimensional vectors, with colored bars indicating different object types. The Temporal Layer depicts processing for a single object type, shared across all objects. The Graph Layer shows processing at one time point, replicated across all time steps.

where $\mathrm{Proj}(\cdot)$ denotes a linear transformation that maps each input variable to a fixed-dimensional vector, ensuring compatibility with subsequent layers.

**Temporal Layer.** To capture temporal dependencies, we employ a unified Transformer-based architecture for all temporal processing layers. Specifically, we apply Transformer encoder layer to the historical node representations across time:

$$h_{i,t}^{\mathrm{temp}} = \mathrm{Transformer}\big(\{h_{i,t-T_{\mathrm{past}}+1}^{\mathrm{node}}, \ldots, h_{i,t}^{\mathrm{node}}\}\big)_t, \tag{4}$$

where the output $h_{i,t}^{\mathrm{temp}}$ denotes the temporally encoded representation of node $v_i$ at time step $t$. To effectively capture temporal dependencies, Transformer-based temporal layers are positioned both between the fusion and graph layers, and following the graph layer. This design enables the model to capture temporal dependencies in the node representations before and after relational aggregation, facilitating deeper modeling of time-evolving dynamics across heterogeneous entities.

**Graph Layer.** We adopt a two-stage relation-aware aggregation strategy tailored for heterogeneous physical graphs. In the first stage, neighbors of the same relation type are aggregated to capture the overall influence of each relation group, avoiding unnecessary complexity in modeling individual interactions. In the second stage, a multi-head graph attention mechanism (GAT) flexibly integrates these relation-specific embeddings, assigning adaptive importance to different relation types for each node. Instead of grouping neighbors merely by node type, we explicitly distinguish edge relations and assign separate parameters per relation. This design balances simplicity and expressiveness: it sufficiently models intra-relation effects while enabling fine-grained, context-aware weighting across relations, avoiding semantic entanglement and excessive parameterization typical of more complex HGNN methods, making it particularly suited for multiphysics systems. At each time step $t$, the system is modeled as a heterogeneous graph $\mathcal{G}_t = (V, E, R)$, where each node aggregates information from its multi-relational neighbors. For each relation type $r_\ell \in R$, we first compute:

$$h_{i,\ell}^{\mathrm{agg}}(t) = \frac{1}{|N_\ell(v_i)|} \sum_{v_j \in N_\ell(v_i)} \mathrm{BiLSTM}_\ell(h_{j,t}^{\mathrm{temp}}), \tag{5}$$

$$\alpha_\ell^k = \mathrm{softmax}\Big(\mathrm{LeakyReLU}\Big(a^{k\top}[W^k h_{i,t}^{\mathrm{temp}} \| W^k h_{i,\ell}^{\mathrm{agg}}(t)]\Big)\Big), \tag{6}$$

$$h_{i,t}^{\mathrm{graph}} = \frac{1}{K} \sum_{k=1}^{K} \sum_{\ell=1}^{L} \alpha_\ell^k W^k h_{i,\ell}^{\mathrm{agg}}(t), \tag{7}$$

where $K$ is the number of attention heads and $L = |R|$ is the number of relation types. $W^k \in \mathbb{R}^{d' \times d}$ is the learnable projection matrix for the $k$-th head, and $a^k \in \mathbb{R}^{2d'}$ is the shared attention vector for computing attention scores. This enables the model to selectively aggregate information from heterogeneous neighbor types in a multi-head attention manner.

**Subtask Model Layer.** To support diverse downstream forecasting objectives across heterogeneous entities, we adopt a modular subtask modeling framework. Each subtask shares a unified decoder architecture that transforms encoded representations into future predictions, as illustrated in Figure 9 in Appendix F.1.

The decoder leverages a masked multi-head attention (MHA) mechanism to align the encoded inputs $\{h_{i,t}\}_{t=t-T_{\text{past}}+1}^{t+T_{\text{future}}}$ with their respective future time steps. The output is then passed through a task-specific dense projection to generate the predicted dynamics $\{\hat{y}_{i,t'}\}_{t'=t+1}^{t+T_{\text{future}}}$:

$$\{\hat{y}_{i,t'}\}_{t'=t+1}^{t+T_{\text{future}}} = \text{Dense}(\text{MHA}(\{h_{i,t}\}_{t=t-T_{\text{past}}+1}^{t+T_{\text{future}}})). \tag{8}$$

To ensure stable information flow and consistent representation, the decoder incorporates Gated Residual Networks (GRNs), gating mechanisms, and Add & Norm layers.

The HGTFT framework has been validated for convergence on a simplified example in Appendix A.1.

## 5 MODEL TRAINING METHODOLOGY

To forecast spatio-temporal dynamics in multi-domain physical systems, we design a progressive HGTFT training pipeline that integrates heterogeneous graph structure, temporal dynamics, and physical constraints. Key components such as multi-instance normalization and physics-informed loss terms ensure stable training and enforce physical consistency, while sequential stages of self-supervised learning, multi-task supervision, and subtask fine-tuning progressively improve generalization and task-specific performance.

### 5.1 MULTI-INSTANCE NORMALIZATION

Normalization is critical for improving model stability, convergence, and generalization. However, standard methods often fall short in our setting due to large intra-type variability (e.g., cooling loads significantly differ by room size) resulting in suboptimal gradient updates and biased loss weighting. To address this, we propose Multi-Instance Normalization. For each variable type $j$ and instance $i$, we compute the time-series min and max values, then aggregate these across instances to derive the $P_{\min}$ percentile of minima and $P_{\max}$ percentile of maxima (e.g., 10th and 90th percentiles). These are used as normalization bounds:

$$\tilde{v}_{i,j}(t) = \frac{v_{i,j}(t) - P_{\min}(\{\min_t v_{i,j}(t)\}_i)}{P_{\max}(\{\max_t v_{i,j}(t)\}_i) - P_{\min}(\{\min_t v_{i,j}(t)\}_i)}. \tag{9}$$

This method ensures consistent scaling across instances of the same object type, improving learning dynamics and overall prediction accuracy. Further comparisons are provided in Appendix H.

### 5.2 SELF-SUPERVISED LEARNING

Training the HGTFT model requires strategies that effectively encode temporal and relational dynamics. Self-supervised learning (SSL) offers a scalable approach by utilizing unlabeled spatio-temporal data via pretext tasks with pseudo-labels. Common tasks such as masked prediction and contrastive learning have demonstrated success in both graph and time-series domains Rani et al. (2023); Xie et al. (2022); Zhang et al. (2024). We formulate SSL as the joint optimization of the foundation model $f_\theta$ and auxiliary heads $p_\phi$ on an unlabeled dataset $D_1$:

$$(\theta^*, \phi^*) = \arg\min_{\theta,\phi} L_{\text{ssl}}(f_\theta, p_\phi, D_1), \tag{10}$$

where $L_{\text{ssl}}$ combines two tasks to capture both temporal dependencies and structural relationships.

**Masked Time-Series Modeling.** Following Zerveas et al. (2021), portions of the input sequence are masked and reconstructed using Mean Squared Error (MSE) loss.

**Masked Edge Modeling.** A subset of graph edges is masked, and the model predicts them via Binary Cross-Entropy (BCE) loss, distinguishing true from randomly sampled negative edges.

To balance the tasks, we use alternating training: the two SSL tasks switch during training, starting and ending with time-series modeling, emphasizing sequence learning while incorporating physical relational understanding. Loss formulations and the SSL training pipeline/results are provided in Appendix E.

## 5.3 Multi-task Supervised Learning

Building on the pre-trained model $f_{\theta^*}$ from SSL, we design a physics-informed multi-task supervised learning (MTSL) framework to fine-tune parameters $\theta^{**}$ with task-specific heads $q_\psi$:

$$(\theta^{**}, \psi^*) = \arg\min_{\theta^*, \psi} L_{\text{MTSL}}(f_{\theta^*}, q_\psi, D_2, Y), \tag{11}$$

where $D_2$ is the labeled spatio-temporal dataset and $Y$ denotes the task labels. Instead of simultaneous MTSL, which scales poorly with task count, we adopt a sequential training strategy that optimizes tasks one-by-one, reducing memory usage and promoting convergence in imbalanced multiphysics settings Vandenhende et al. (2021); Yu et al. (2024).

**Convergence Criterion.** Sequential training is considered converged when the average relative change in task losses falls below a threshold. Formally, for task $i$ at iteration $k$:

$$\Delta L_{\text{task},i}^{(k)} = \frac{|L_{\text{task},i}^{(k)} - L_{\text{task},i}^{(k-1)}|}{L_{\text{task},i}^{(k-1)}}, \quad \Delta L_{\text{avg}}^{(k)} = \frac{1}{N} \sum_{i=1}^{N} \Delta L_{\text{task},i}^{(k)}, \tag{12}$$

where $N$ is the number of tasks. Convergence is reached when $\Delta L_{\text{avg}}^{(k)}$ falls below a predefined threshold (e.g., 2%).

**Physics-informed Loss Design.** To embed physical consistency directly into model training, we augment the standard MSE loss with three domain-informed components: (1) Reasonableness Checks Score (RCS) discourages predictions that violate operational constraints or physical laws Appendix F.4; (2) Correlation-Based Score (CRS) promotes consistency with known correlations in multivariate time-series data Appendix F.5; (3) Frequency Domain Similarity (FDS) aligns predicted and actual spectral characteristics Appendix F.6. The total loss for each task is the weighted sum of the four loss components, with learnable or pre-defined weights. We adopt a hard parameter sharing scheme with a shared encoder and task-specific decoders, enabling the model to generalize across tasks while retaining task-specific specialization. The weighting scheme and training settings for different stages are detailed in Appendix F.7.

## 5.4 Subtask Fine-tuning

The subtask fine-tuning process consists of two stages: task fine-tuning and project-specific fine-tuning. **Task fine-tuning** adapts the pre-trained model to forecasting tasks by freezing shared encoder layers and updating only task-specific parameters, enhancing performance and serving as pre-adaptation. **Project-specific fine-tuning** adapts the model to real-world scenarios with limited labels, updating only lightweight components (e.g., dense projection) to align with new data while retaining general representations from pretraining.

## 6 Experiments

### 6.1 Datasets

**Standard and Graph-Structured Datasets.** Common benchmarks for time-series forecasting fall into two categories. The first includes standard datasets such as ETT, Weather, and Electricity Haixu et al. (2022), which assess general temporal prediction under purely data-driven assumptions. The second includes graph-structured datasets such as PeMSD4, PeMSD8 Chen et al. (2001),

and COVID-19 case data Dong et al. (2020); nyt, where each node has a time series and spatial dependencies are encoded in graphs. While valuable for studying spatiotemporal correlations, these datasets do not capture the complexity of multi-domain physical systems considered in this work.

**Multi-domain physical System Datasets.** Energy and building operations provide a more representative scenario for multi-domain physics forecasting. Building systems comprise diverse components governed by distinct physical mechanisms: rotational and flow devices (pumps, compressors, valves), heat exchange units (fan coils, radiators, exchangers), transport infrastructures (pipes, ducts, tanks), and sensing/control units (thermostats, flow meters, controllers). These interact through principles of heat transfer, fluid dynamics, thermodynamics, and mass/energy conservation. The diversity and interdependence of such components make building systems a meaningful and broadly applicable testbed for multi-domain physical forecasting.

We first include the Building Time-Series (BTS) dataset, recently released at NeurIPS 2024 Prabowo et al. (2024), which contains over ten thousand time-series variables collected from three real buildings over a three-year period, covering hundreds of unique ontologies. While valuable, its scale remains limited for comprehensive pre-training. We release the Multiphysics Building System (MBS) dataset, which combines real-world and simulated building data. Further details and access to the dataset via an anonymous link are provided in Appendix B.

## 6.2 BASELINES

We compare our approach against a diverse set of baselines, encompassing traditional machine learning models, graph-based methods, and recent advancements in large pre-trained time-series models. Traditional models such as LSTM Hochreiter (1997), as well as more recent architectures like Autoformer Wu et al. (2021), forecast each variable independently without incorporating structural information. TFT Lim et al. (2021) integrates static covariates with dynamic time-series inputs for multivariate forecasting. HTGNN Fan et al. (2022) and STD-MAE Gao et al. (2023) operate on graph-structured time-series data, with heterogeneous and homogeneous structures, respectively. Recent developments in large pre-trained models have shown significant promise. TimesFM Das et al. (2023) and MOIRAI Woo et al. (2024) represent general pre-trained time-series models. LLM-based approaches, including Time-LLM Jin et al. (2023) and LLMTimed Gruver et al. (2024), leverage large language models for time-series prediction.

## 6.3 MAIN RESULTS

We first assess HGTFT on graph-structured spatiotemporal datasets, where relational information is explicit but no physical constraints are provided. This setting evaluates general forecasting ability against data-driven baselines. As shown in Table 1, HGTFT consistently ranks top-2 on PeMSD4, PeMSD8, and COVID-19, confirming its strength in capturing structured relationships. In particular, the COVID-19 dataset exhibits complex, multi-scale dynamics driven by non-stationary interventions and heterogeneous regional attributes. Unlike more stable traffic datasets, it tests a model's ability to capture diverse entities and their interactions, where HGTFT demonstrates clear advantages. For completeness, we also evaluated HGTFT on standard time-series benchmarks (e.g., ETT) in Appendix G, which are less aligned with the problem studied here.

Table 1: Performance on spatiotemporal datasets. COVID-19 (JHU): daily infection counts from 83 Michigan counties; COVID-19 (NYT): daily death counts from 50 U.S. states. All models are trained or fine-tuned on 10% of each dataset. Best results are in **bold**, second best are underlined.

| Dataset | Metric | LSTM | Autoformer | TFT | HTGNN | STD-MAE | TimesFM | MOIRAI | LLMTime | Time-LLM | HGTFT (Ours) |
|---|---|---|---|---|---|---|---|---|---|---|---|
| PeMSD4 | MAE | 32.48 | 32.39 | 31.32 | 21.01 | **17.85** | 32.57 | 33.31 | 33.69 | 32.23 | 19.94 |
| | RMSE | 55.51 | 53.19 | 48.37 | 36.44 | **29.72** | 55.94 | 55.51 | 52.49 | 52.18 | 32.16 |
| PeMSD8 | MAE | 24.98 | 25.56 | 24.63 | 18.22 | **13.67** | 23.93 | 24.03 | 26.68 | 27.74 | 16.43 |
| | RMSE | 41.71 | 41.65 | 39.74 | 27.04 | **22.62** | 42.41 | 42.49 | 43.94 | 40.01 | 25.08 |
| COVID-19 (JHU) | MAE | 122.42 | 115.77 | 121.81 | 46.24 | 47.75 | 99.75 | 105.74 | 115.37 | 95.01 | **41.54** |
| | RMSE | 232.11 | 198.67 | 261.77 | 102.73 | **92.62** | 216.63 | 234.24 | 216.74 | 201.14 | 94.38 |
| COVID-19 (NYT) | MAE | 70.59 | 62.37 | 71.36 | 31.16 | 26.69 | 57.07 | 81.05 | 72.24 | 83.17 | **25.69** |
| | RMSE | 139.18 | 133.35 | 158.93 | 75.98 | 72.98 | 113.45 | 134.57 | 157.31 | 146.45 | **65.64** |

We evaluate HGTFT on the open BTS dataset, which includes three anonymized buildings Prabowo et al. (2024), as a representative multi-domain physical system. Forecasting tasks use the previous 7 days (672 time steps) to predict the next day (96 time steps) at 15-minute intervals, with all metrics computed on normalized values to account for inter-variable scale differences. We follow three evaluation settings: (i) pretraining on 50 randomly selected MBS buildings, (ii) pretraining on the full MBS dataset, and (iii) direct training on 30 days of each BTS building's data with the remaining days for evaluation. For the pretrained models, both zero-shot prediction (without BTS building-specific data) and few-shot adaptation (using 30 days of BTS data) are assessed. Settings (i) and (ii) leverage the training methodology and physics-informed losses introduced earlier. The experiments are repeated 10 times with different seeds for pretraining building selection and few-shot sampling, and results are averaged.

As shown in Table 2, HGTFT achieves strong zero-shot performance, further enhanced by few-shot adaptation. Even without pretraining and physics-informed losses, it consistently surpasses all baselines, reducing MSE by up to 38% and RCS by 25% relative to the second-best model. Physics-informed pretraining yields an order-of-magnitude improvement in RCS, demonstrating its effectiveness in enforcing physical consistency. Few-shot adaptation substantially lowers MSE, while keeping RCS only slightly higher yet still well-controlled, striking a balance between predictive accuracy and physical plausibility. Although the benefits of physics-informed pretraining are limited for purely temporal models, they extend to spatial–temporal approaches such as HTGNN and STD-MAE, highlighting the broader generalization potential of physics-aware training. The strongest gains are observed for HGTFT, reflecting its capacity to integrate heterogeneous dynamics with structured physical constraints.

Table 2: Time-series forecasting results on the BTS dataset under three settings: pretrained zero-shot, pretrained few-shot, and no pre-training. Best results are in **bold**, second-best are underlined.

| Settings | Metric | LSTM | Autoformer | TFT | HTGNN | STD-MAE | TimesFM | MOIRAI | LLMTime | Time-LLM | HGTFT (Ours) |
|---|---|---|---|---|---|---|---|---|---|---|---|
| Zero-shot (50 MBS) | MSE | 0.0142 | 0.0136 | 0.0105 | _0.0091_ | 0.0095 | 0.0163 | 0.0159 | 0.0232 | 0.0220 | **0.0056** |
| | RCS | 0.0903 | 0.0567 | 0.0321 | _0.0179_ | 0.0197 | 0.0696 | 0.0507 | 0.0733 | 0.0786 | **0.0025** |
| | CRS | 0.9687 | 1.0019 | 1.1071 | 0.7302 | _0.4494_ | 0.8852 | 1.0398 | 1.1942 | 1.1813 | **0.4437** |
| | FDS | 0.8347 | 0.9994 | 0.9747 | 1.0049 | 0.9038 | 0.8135 | _0.6660_ | 1.2103 | 0.9644 | **0.5745** |
| Few-shot (50 MBS) | MSE | 0.0082 | 0.0064 | _0.0062_ | 0.0063 | 0.0070 | 0.0083 | 0.0086 | 0.0120 | 0.0139 | **0.0036** |
| | RCS | 0.0750 | 0.0588 | 0.0281 | 0.0297 | _0.0277_ | 0.0528 | 0.0458 | 0.0601 | 0.0556 | **0.0037** |
| | CRS | 0.7878 | 0.8703 | 0.8121 | 0.6518 | **0.4274** | 0.8937 | 0.8537 | 0.8695 | 0.8809 | _0.5132_ |
| | FDS | 0.6062 | 0.5697 | 0.5936 | 0.5760 | 0.6780 | _0.4943_ | 0.5085 | 0.7417 | 0.6437 | **0.4303** |
| Zero-shot (Full MBS) | MSE | 0.0133 | 0.0122 | 0.0107 | 0.0087 | _0.0084_ | 0.0123 | 0.0179 | 0.0219 | 0.0181 | **0.0047** |
| | RCS | 0.0747 | 0.0514 | 0.0326 | _0.0158_ | 0.0194 | 0.0645 | 0.0517 | 0.0704 | 0.0803 | **0.0018** |
| | CRS | 1.0924 | 1.1426 | 0.8729 | 0.7826 | _0.6086_ | 1.0770 | 1.1654 | 1.2520 | 1.0313 | **0.4472** |
| | FDS | 0.7306 | 0.8617 | 0.8825 | 0.8798 | 1.1247 | 0.8398 | _0.7114_ | 1.2927 | 1.0047 | **0.5045** |
| Few-shot (Full MBS) | MSE | 0.0073 | 0.0072 | _0.0056_ | 0.0059 | 0.0065 | 0.0084 | 0.0087 | 0.0128 | 0.0112 | **0.0033** |
| | RCS | 0.0744 | 0.0553 | 0.0300 | _0.0244_ | 0.0291 | 0.0554 | 0.0452 | 0.0600 | 0.0571 | **0.0032** |
| | CRS | 0.7383 | 0.8252 | 0.7387 | 0.6035 | _0.4107_ | 0.7636 | 0.7942 | 0.8383 | 0.9016 | **0.4083** |
| | FDS | 0.5419 | 0.6185 | 0.5177 | 0.5945 | 0.6603 | **0.4495** | 0.4986 | 0.7663 | 0.6331 | _0.4691_ |
| No Pretrain | MSE | 0.0082 | 0.0075 | _0.0062_ | 0.0064 | 0.0072 | 0.0092 | 0.0086 | 0.0127 | 0.0113 | **0.0040** |
| | RCS | 0.0697 | 0.0537 | _0.0278_ | 0.0315 | 0.0377 | 0.0560 | 0.0440 | 0.0596 | 0.0591 | **0.0209** |
| | CRS | 0.7378 | 0.8103 | 0.7110 | 0.6079 | **0.4271** | 0.8108 | 0.8096 | 0.8624 | 0.8215 | _0.5814_ |
| | FDS | 0.6313 | 0.6399 | 0.5814 | 0.7077 | 0.6490 | _0.5279_ | **0.4957** | 0.8392 | 0.6683 | 0.6255 |

## 6.4 ABLATION STUDY

To contextualize the ablation results, we additionally include baseline models on the MBS dataset, providing a horizontal reference for interpreting the performance levels before and after ablating each component. This allows us to more clearly quantify the contribution of static–dynamic fusion, structural graph modeling, temporal modules, and the subtask layer under a consistent evaluation setup.

We then ablate major design choices of HGTFT. For static–dynamic fusion, VSNs are replaced with dense layers. For structural modeling, we remove the graph encoder or substitute GAT-based aggregation. For temporal modeling, we vary Transformer depth, place a single layer before or after the graph layer (Pre-G/Post-G), or remove it. The subtask layer is simplified by removing GRU and Add & Norm units or retaining only a dense projection. Results on MSE, RCS, CRS, and FDS are

reported in Table 4. Model scaling (Appendix D) further shows performance improves with size up to 310M parameters, beyond which gains plateau, suggesting 310M as an efficient capacity balance.

Table 3: Average forecasting performance over 10 runs on 50 randomly selected building cases from the MBS dataset. Best results in **bold**, second best underlined.

| Metric | LSTM | Autoformer | TFT | HTGNN | STD-MAE | TimesFM | MOIRAI | LLMTime | Time-LLM | HGTFT zero-shot | HGTFT few-shot |
|---|---|---|---|---|---|---|---|---|---|---|---|
| MSE | 0.0049 | 0.0053 | 0.0044 | 0.0048 | 0.0051 | 0.0064 | 0.0072 | 0.0093 | 0.0085 | 0.0027 | **0.0023** |
| RCS | 0.0376 | 0.0298 | 0.0152 | 0.0133 | 0.0206 | 0.0287 | 0.0243 | 0.0320 | 0.0307 | **0.0012** | 0.0029 |
| CRS | 0.4763 | 0.5207 | 0.5028 | 0.3855 | 0.2804 | 0.5311 | 0.5258 | 0.5527 | 0.5694 | 0.3123 | **0.2581** |
| FDS | 0.4382 | 0.4547 | 0.4288 | 0.4761 | 0.4930 | 0.3684 | 0.3899 | 0.5781 | 0.5139 | 0.4052 | **0.2919** |

Table 4: Ablation results on architecture modifications and simplifications on the MBS dataset.

| Metric | HGTFT | Fusion: dense | Graph: removed | Graph: GAT | Temporal: removed | Temporal: Pre-G | Temporal: Post-G | Subtask: w/o GRU | Subtask: dense |
|---|---|---|---|---|---|---|---|---|---|
| MSE | **0.0027** | 0.0053 | 0.0065 | 0.0032 | 0.0072 | 0.0063 | 0.0064 | 0.0048 | 0.0067 |
| RCS | **0.0012** | 0.0247 | 0.0343 | 0.0037 | 0.0157 | 0.0112 | 0.0103 | 0.0136 | 0.0297 |
| CRS | **0.3123** | 0.5229 | 0.3551 | 0.3377 | 0.6324 | 0.5363 | 0.5289 | 0.4622 | 0.5435 |
| FDS | **0.4052** | 0.4174 | 0.4139 | 0.4158 | 0.5890 | 0.5118 | 0.5152 | 0.4961 | 0.5803 |

## 6.5 FURTHER ANALYSIS

To evaluate the model's capacity to capture multiphysics interactions, Figure 2 visualizes predicted temperature fields on a sample floor at a selected time slice. The HGTFT model using full inputs (dynamic, static, and graph data) accurately reconstructs spatial temperature patterns. In contrast, the variant excluding static features (e.g., zone type, orientation) and spatial adjacency yields less coherent results, underscoring the importance of incorporating static and graph information.

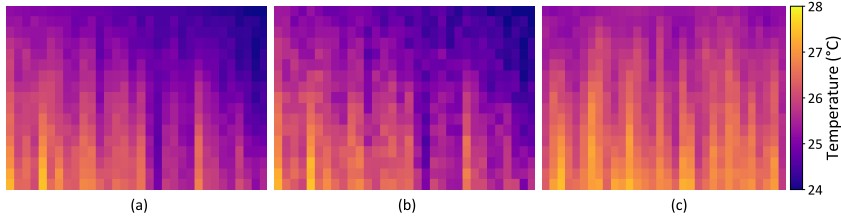

Figure 2: Predicted temperature fields for a sample floor at a selected time slice. (a) ground truth, (b) prediction by HGTFT using full input, and (c) prediction without static or graph information.

To assess the impact of physics-aligned learning, Figure 3 presents model responses under low-frequency control changes, where the number of chillers increases from 2 to 3 and then to 4. The physics-aware HGTFT (solid lines) generates trends consistent with thermodynamic principles: adding chillers raises chilled water flow while reducing indoor temperature and humidity. In contrast, the MSE-only baseline (dashed lines) exhibits muted responses, with clustered curves that fail to capture the expected physical effects. These results demonstrate that incorporating physics-aligned supervision not only improves generalization to rare control actions but also enforces physically consistent predictions. Metric ablation further supports this conclusion: removing RCS, CRS, or FDS degrades their corresponding scores from 0.0012 to 0.0134 (RCS), 0.312 to 0.384 (CRS), and 0.405 to 0.423 (FDS), while leaving other metrics largely unchanged. Among them, RCS proves to be the most influential.

To further validate our methodology, we conduct extensive analyses. First, we compare normalization strategies using CV-RMSE across variable types and observe that the proposed Multi-Instance Normalization consistently improves optimization stability and generalization over Min-Max and

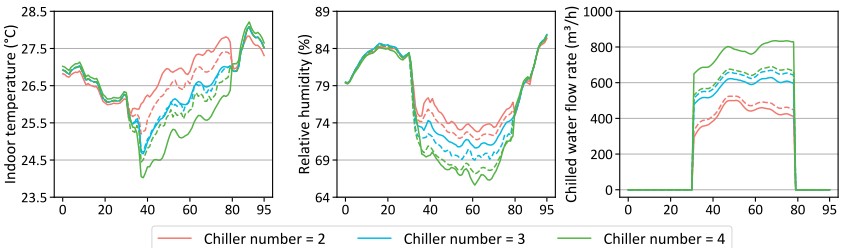

Figure 3: Predicted system responses to changes in control actions (solid: multi-metric training model; dashed: MSE-only training model).

Z-Score methods (Appendix H). For self-supervised learning, we evaluate three training strategies and find that while simultaneous task training impairs forecasting quality, our alternating task training method effectively balances time-series and relational representations with lower losses (Appendix J.1). In supervised learning, sequential multi-task training yields stable convergence, and reducing either task types or data coverage leads to moderate performance drops, highlighting the importance of task and case diversity (Appendix J.2). Input/output horizon analysis reveals that longer input patches enhance short- to mid-term forecasting accuracy, demonstrating the value of extended temporal context (Appendix J.3).

## 7 CONCLUSION

This paper addresses time series forecasting in heterogeneous multi-domain physical systems, where diverse entities, relations, and variables interact under physical constraints. We introduce the HGTFT, which integrates heterogeneous tokenization, graph-temporal fusion, and physics-aligned supervision within a pre-training and fine-tuning paradigm. Experiments show that HGTFT not only achieves performance comparable to state-of-the-art models on multiple spatiotemporal benchmarks, but also delivers clear advantages in realistic multiphysics scenarios, with strong zero-shot generalization and further gains through few-shot adaptation. These results highlight HGTFT as a robust and scalable framework for forecasting in complex physical environments. Limitations and future work are discussed in Appendix K.

## 8 ETHICS STATEMENT

This work adheres to the ICLR Code of Ethics. The study does not involve human subjects, private data, or personally identifiable information. All datasets used are publicly available, and additional processed datasets are shared through an anonymized link in the supplementary materials to ensure fair and ethical access. The methods and results do not pose foreseeable risks of discrimination, unfair bias, or harmful applications. To the best of our knowledge, this work complies with standards of research integrity, legal requirements, and ethical scientific conduct.

## 9 REPRODUCIBILITY STATEMENT

We have made significant efforts to ensure reproducibility. The model architecture and training methodology are provided in Section 4 and Section 5 of the main text, with additional implementation and reproducibility details presented in Appendix E and Appendix F. The theoretical foundations of the proposed framework are formally validated in Appendix A. The implementation code is submitted as supplementary materials with clear instructions. All primary datasets employed in this study are publicly accessible. Furthermore, parts of our supplementary datasets are released through an anonymized link (`https://drive.google.com/drive/folders/1fOG6SdFXXdJ0LtaELQA6o7obRxgTBfpg?usp=sharing`) to facilitate independent validation.

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

## A  DESCRIPTION OF TYPICAL PROBLEMS

We present two examples to highlight the significance of extending the problem within the context of multiphysics systems (e.g., building systems). The first example involves a relatively simple dynamic system model, which begins with the fan coil unit (FCU) in relation to the space and cooling source. We then extend to a more complex system, which includes multiple types of objects and relationships, with each type of object potentially having a large number of instances.

### A.1  EXAMPLE 1: HEAT EXCHANGE IN FAN COIL UNIT (FCU)

In the FCU, heat exchange occurs between air and water, and this process can be modeled using differential equations. Let's define the problem and derive the equations step by step.

**Problem Definition**

The heat exchange process involves the flow of air and water through the FCU, where air absorbs heat from the water and vice versa. The temperature dynamics for air and water are described as follows:

We first describe the air temperature dynamics. The rate of change of air temperature is governed by the following equation:

$$\dot{m}_{\mathrm{air}} c_{\mathrm{air}} \frac{dT_{\mathrm{air}}(t)}{dt} = \dot{m}_{\mathrm{air}} c_{\mathrm{air}} (T_{\mathrm{in,air}}(t) - T_{\mathrm{out,air}}(t)) - Q_{\mathrm{heat,air}}(t) \tag{13}$$

where: - $\dot{m}_{\mathrm{air}}$ is the mass flow rate of air (kg/s), - $c_{\mathrm{air}}$ is the specific heat capacity of air (J/kg·K), - $T_{\mathrm{in,air}}(t)$ and $T_{\mathrm{out,air}}(t)$ are the inlet and outlet air temperatures at time $t$ (°C or K), - $Q_{\mathrm{heat,air}}(t)$ is the heat exchanged between air and water at time $t$ (W).

Next, we consider the water temperature dynamics. The change in water temperature over time can be described as:

$$\dot{m}_{\mathrm{water}} c_{\mathrm{water}} \frac{dT_{\mathrm{water}}(t)}{dt} = Q_{\mathrm{heat,water}}(t) - Q_{\mathrm{water,out}}(t) \tag{14}$$

where: - $\dot{m}_{\mathrm{water}}$ is the mass flow rate of water (kg/s), - $c_{\mathrm{water}}$ is the specific heat capacity of water (J/kg·K), - $T_{\mathrm{water}}(t)$ is the water temperature at time $t$ (°C or K), - $Q_{\mathrm{heat,water}}(t)$ is the heat exchanged between air and water at time $t$ (W), - $Q_{\mathrm{water,out}}(t)$ is the heat lost by water to external factors at time $t$ (W).

The heat exchange between air and water is modeled by the following equation:

$$Q_{\mathrm{heat,air}}(t) = Q_{\mathrm{heat,water}}(t) = h_{\mathrm{air-water}} A_{\mathrm{heat}} (T_{\mathrm{air}}(t) - T_{\mathrm{water}}(t)) \tag{15}$$

where: - $h_{\mathrm{air-water}}$ is the heat transfer coefficient between air and water (W/m²·K), - $A_{\mathrm{heat}}$ is the heat exchange area (m²), - $T_{\mathrm{air}}(t)$ and $T_{\mathrm{water}}(t)$ are the air and water temperatures at time $t$ (K).

**Derivation of Differential Equations**

Combining the heat exchange formulas with the temperature dynamics, we get a system of differential equations:

$$\dot{m}_{\mathrm{air}} c_{\mathrm{air}} \frac{dT_{\mathrm{air}}(t)}{dt} = \dot{m}_{\mathrm{air}} c_{\mathrm{air}} (T_{\mathrm{in,air}}(t) - T_{\mathrm{out,air}}(t)) - h_{\mathrm{air-water}} A_{\mathrm{heat}} (T_{\mathrm{air}}(t) - T_{\mathrm{water}}(t)) \tag{16}$$

$$\dot{m}_{\mathrm{water}} c_{\mathrm{water}} \frac{dT_{\mathrm{water}}(t)}{dt} = h_{\mathrm{air-water}} A_{\mathrm{heat}} (T_{\mathrm{air}}(t) - T_{\mathrm{water}}(t)) - Q_{\mathrm{water,out}}(t) \tag{17}$$

**Introducing Temperature Difference**

To simplify the equations, introduce the temperature difference:

$$\Delta T(t) = T_{\text{air}}(t) - T_{\text{water}}(t) \tag{18}$$

Thus, the air temperature and water temperature can be expressed as:

$$T_{\text{air}}(t) = T_{\text{water}}(t) + \Delta T(t) \tag{19}$$

Substituting this into the differential equations, we get:

For the air temperature equation:

$$\dot{m}_{\text{air}} c_{\text{air}} \frac{d\Delta T(t)}{dt} = \dot{m}_{\text{air}} c_{\text{air}} (T_{\text{in,air}}(t) - T_{\text{out,air}}(t)) - h_{\text{air}-\text{water}} A_{\text{heat}} \Delta T(t) \tag{20}$$

For the water temperature equation:

$$\dot{m}_{\text{water}} c_{\text{water}} \frac{dT_{\text{water}}(t)}{dt} = h_{\text{air}-\text{water}} A_{\text{heat}} \Delta T(t) - Q_{\text{water,out}}(t) \tag{21}$$

**Analytical Solution**

For the temperature difference equation $\Delta T(t)$, we obtain:

$$\dot{m}_{\text{air}} c_{\text{air}} \frac{d\Delta T(t)}{dt} = \dot{m}_{\text{air}} c_{\text{air}} (T_{\text{in,air}}(t) - T_{\text{out,air}}(t)) - h_{\text{air}-\text{water}} A_{\text{heat}} \Delta T(t) \tag{22}$$

This is a first-order linear differential equation, which can be solved as:

$$\Delta T(t) = \frac{\dot{m}_{\text{air}} c_{\text{air}} (T_{\text{in,air}}(t) - T_{\text{out,air}}(t))}{h_{\text{air}-\text{water}} A_{\text{heat}}} \left( 1 - e^{-\frac{h_{\text{air}-\text{water}} A_{\text{heat}}}{\dot{m}_{\text{air}} c_{\text{air}}} t} \right) \tag{23}$$

Using the initial condition $\Delta T_0 = T_{\text{air},0} - T_{\text{water},0}$, we obtain:

$$T_{\text{air}}(t) = T_{\text{water}}(t) + \frac{\dot{m}_{\text{air}} c_{\text{air}} (T_{\text{in,air}}(t) - T_{\text{out,air}}(t))}{h_{\text{air}-\text{water}} A_{\text{heat}}} \left( 1 - e^{-\frac{h_{\text{air}-\text{water}} A_{\text{heat}}}{\dot{m}_{\text{air}} c_{\text{air}}} t} \right) \tag{24}$$

The analytical solution for the water temperature is:

$$T_{\text{water}}(t) = T_{\text{water},0} + \frac{h_{\text{air}-\text{water}} A_{\text{heat}} \Delta T_0}{\dot{m}_{\text{water}} c_{\text{water}}} \left( 1 - e^{-\frac{h_{\text{air}-\text{water}} A_{\text{heat}}}{\dot{m}_{\text{air}} c_{\text{air}}} t} \right) - \frac{Q_{\text{water,out}}(t)}{\dot{m}_{\text{water}} c_{\text{water}}} \tag{25}$$

**Challenges and Complexities**

The heat lost by water, $Q_{\text{water,out}}$, is influenced by heat/cooling source objects, while $T_{\text{in,air}}$ and $T_{\text{out,air}}$ are connected to spaces. Both heat/cooling source objects and spaces have their own distinct features and dynamics. The primary challenge in modeling such a system lies in the complex coupling of air and water dynamics, as well as the interactions between multiple spaces and Fan Coil Units (FCUs). As the number of spaces and FCUs increases, the complexity of the system grows exponentially, making it increasingly difficult to derive a closed-form solution. Therefore, the ability to integrate multiple object types and relationships through neural network algorithms is a critical requirement for addressing such problems.

**Numerical Simulation and Prediction Using HGTFT**

In this study, we assign different values to the static parameters in the previously defined mathematical model and apply time-varying functions to the external variables, $Q_{\text{water,out}}(t)$, $T_{\text{in,air}}(t)$, and $T_{\text{out,air}}(t)$. Through numerical simulations, a dataset is generated, which is then used to train the HGTFT-based model. The objective of this training is to predict the temperature profiles $T_{\text{water}}(t)$ and $T_{\text{air}}(t)$ based on the temporal variations of the external variables and the given static parameters.

Figure 4 illustrates a numerical simulation where time-series values are generated under the assumption of sinusoidal variations for $Q_{\mathrm{water,out}}(t)$, $T_{\mathrm{in,air}}(t)$, and $T_{\mathrm{out,air}}(t)$. The figure also shows the corresponding predictions of $T_{\mathrm{water}}(t)$ and $T_{\mathrm{air}}(t)$ obtained using the HGTFT model.

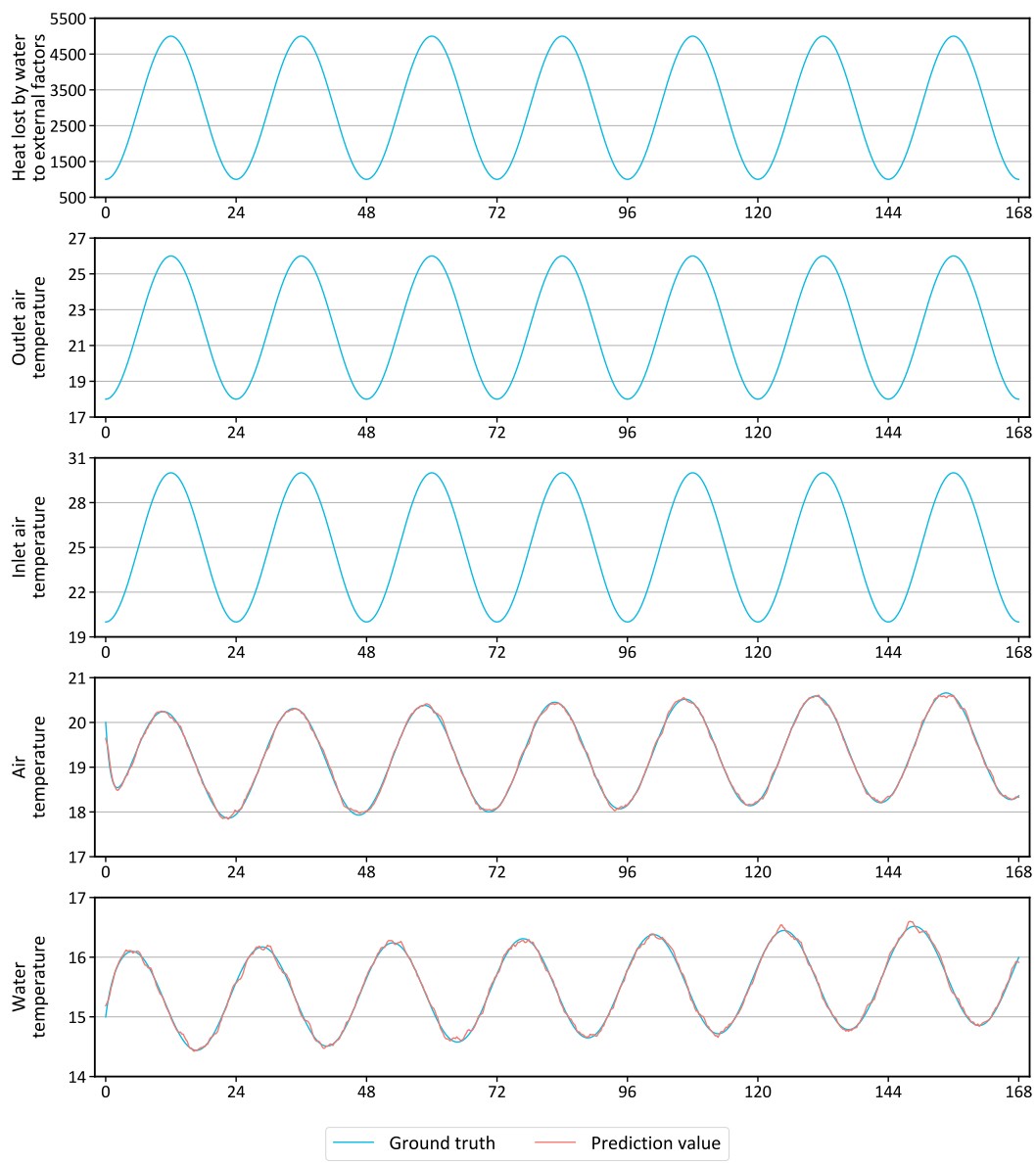

Figure 4: Numerical simulation for Example 1 and predictions using the HGTFT model.

## A.2 EXAMPLE 2: FORECASTING IN A HETEROGENEOUS MULTIPHYSICS HVAC NETWORK

This example highlights a forecasting task in a complex HVAC operation scenario involving heterogeneous entities with interconnected physical relationships. As shown in Table 5, the task spans multiple object types—such as environmental sensors, thermal zones, chillers, pumps, and air-side equipment—each with distinct static attributes and time-dependent dynamics. The temporal forecasting goal varies across objects, with certain variables provided as future inputs (e.g., control signals or setpoints), while others are to be predicted. This setup reflects real-world complexity where forecasting depends on both physical coupling (e.g., energy and fluid flow) and system control behavior.

Table 5: An example of an HVAC operation task, covering multiple object types—such as environment, general zone, chiller, chilled water pump (ACCCCP), cooling water pump (ACCCOP), cooling tower (ACCCOT), fan coil unit (ACATFC), and supply air fan (ACATFU)—along with their associated input and output variable types.

| Object type | Input | | | Output |
|---|---|---|---|---|
| | Static attribute | Dynamic variable for the past | Dynamic attribute for the future | Dynamic attribute for the future |
| Environment | | Outdoor temperature | Outdoor temperature | |
| GeneralZone | Area, volume, orientation | Indoor temperature, relative humidity | | Indoor temperature, relative humidity |
| Chiller | Rated cooling capacity, rated power | Chilled water supply temperature, chilled water return temperature, chilled water flow rate | | Chilled water supply temperature, chilled water return temperature, chilled water flow rate |
| ACCCCP | Rated power, rated flow rate, rated head | Operating status, operating power, flow rate | Operating status | Operating power, flow rate |
| ACCCOP | Rated power, rated flow rate, rated head | Operating status, operating power, flow rate | Operating status | Operating power, flow rate |
| ACCCOT | Rated power, rated air flow, number of fans, design outdoor wet-bulb temperature | Number of operating fans, air flow rate, leaving water temperature, water flow rate, leaving water temperature setpoint | Leaving water temperature Setpoint | Number of Operating fans, air flow rate, leaving water temperature, water flow rate |
| ACATFC | Rated power, rated air flow, rated chilled water flow rate | Supply air temperature, return air temperature, supply air temperature setpoint | Supply air temperature setpoint | Supply air temperature, return air temperature |
| ACATFU | Rated power, rated air flow | Fresh air flow rate, fan speed | Fan speed | Fresh air flow rate |

# B  MBS DATASET DETAILS

The Multi-physics Building System (MBS) dataset combines real-world and simulated building data. A subset is publicly available at `https://drive.google.com/drive/folders/1fOG6SdFXXdJ0LtaELQA6o7obRxgTBfpg?usp=sharing`. Object and relationship definitions in building operation systems are based on a standardized, publicly available data dictionary commonly used in building automation. The training dataset primarily contains HVAC-related data, including empirical data aggregated from diverse real-world deployments and synthetic data generated via a high-fidelity simulation environment. Figure 5 shows a partial 3D visualization from the simulation setup, illustrating mappings between equipment and spatial zones, as well as detailed interconnections such as piping and ductwork.

## B.1  REAL PROJECT DATA

We have accumulated a substantial dataset from a multitude of real-world projects, encompassing various building subsystems such as energy management systems, security surveillance systems, equipment and facility management systems, and building automation systems. The dataset comprises a total of 1045 projects, with 508 projects containing relatively comprehensive information. The dataset contains about 5B tokens and 16B time points data.

## B.2  SIMULATION DATA

Compared to real-world project data, simulations can involve a much larger number of variables, including many that are difficult or even impossible to measure in the real world project but can be calculated in a simulation environment. Additionally, simulations allow for the alteration of many operating conditions, covering a much broader range of scenarios than real projects can achieve. Given the astronomical number of possible parameter combinations, it is necessary to reduce the number of generated simulation cases. This can be achieved by carefully selecting variable parameters and applying orthogonal testing to optimize the case generation process. We constructed a massive dataset of building energy simulations using EnergyPlus DOE (2015). By systematically varying key building parameters across 12 diverse base building models, we generated approxi-

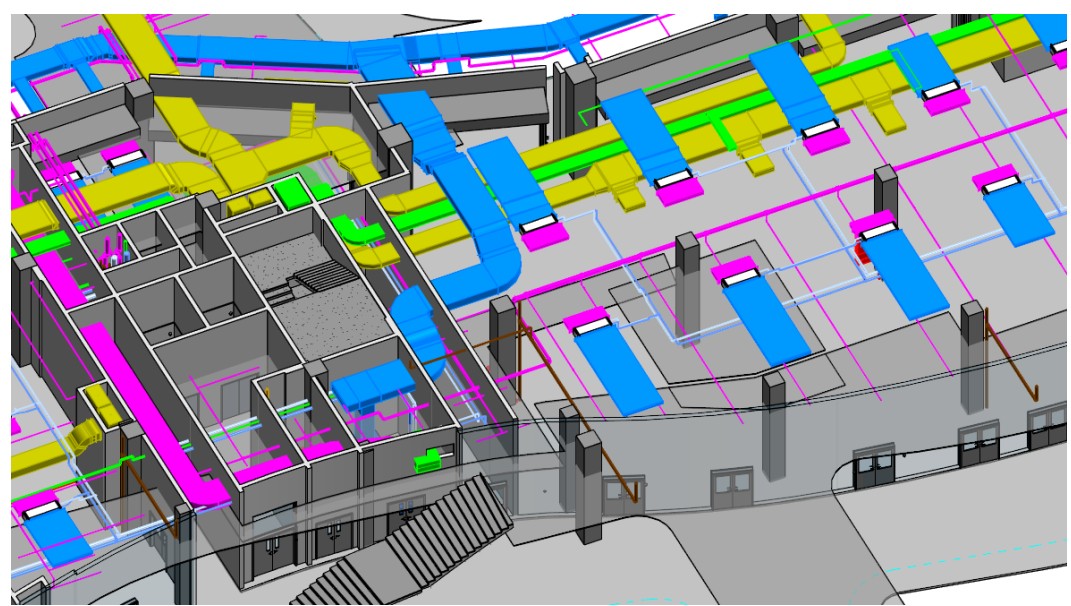

Figure 5: 3D illustration of a simulated building environment, showing spatial layout, service relationships between equipment and zones, and extensive duct and piping connections representing air and water flows in HVAC systems.

mately 5,000 simulation scenario cases. Each case provides high-resolution 15-minute data for a year, resulting in a dataset of over 80B tokens and 600B time points data.

### B.3 COMPARISON BETWEEN REAL PROJECT AND SIMULATION DATA

We collected both simulated and real-world data for various variables, and Figure 6 illustrates a daily profile of chiller plant cooling power, for instance. Overall, the simulated data closely aligns with the real-world data, demonstrating a strong consistency. Due to the ability to simulate a wider range of operating conditions, the simulated data offers a broader coverage of scenarios. This increased diversity in the simulated conditions allows for a more comprehensive representation of potential system behaviors, enhancing the robustness of the model training and its ability to generalize to different operational contexts.

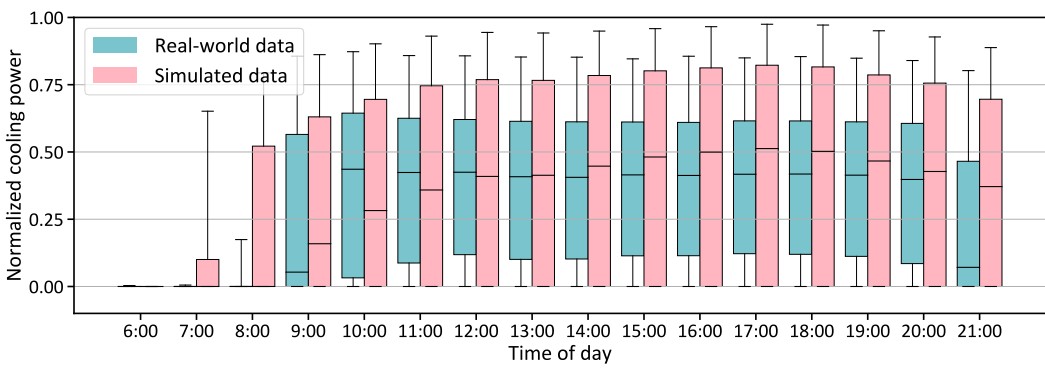

Figure 6: Comparison of a daily profile for chiller plant cooling power between real-world data and simulated data.

# C SUPPLEMENTARY EXPLANATION OF NETWORK UNITS AND FORMULAS

## C.1 GATED RESIDUAL NETWORK (GRN)

The following description of the Gated Residual Network (GRN) is primarily based on the relevant sections from the Temporal Fusion Transformer (TFT) paper Lim et al. (2021). The GRN introduces a gating mechanism via the Gated Linear Unit (GLU) to regulate the flow of information and selectively pass only the most relevant inputs. This design is critical for handling diverse data inputs effectively. The GRN structure is described by Equations 26-28. The primary input $a$ and the context input $c$ are processed through the Exponential Linear Unit (ELU) activation function, linear transformation, GLU, and layer normalization. Weight matrices $W_1$, $W_2$, $W_3$, and biases $b_1$, $b_2$ govern the transformation, providing flexibility through selective non-linear processing.

$$\text{GRN}(a, c) = \text{LayerNorm}(a + \text{GLU}(\eta_1)), \tag{26}$$

$$\eta_1 = W_1 \eta_2 + b_1, \tag{27}$$

$$\eta_2 = \text{ELU}(W_2 a + W_3 c + b_2). \tag{28}$$

The GLU is defined in Equation 29, where $X$ is the input, $W_4$ and $W_5$ are learnable weights, $b_3$ and $b_4$ are biases, and $\sigma$ is the sigmoid function. The Hadamard product $\odot$ modulates the GRN's influence on the input $a$, allowing it to potentially skip processing when the GLU output approaches zero. If no context vector is provided, $c$ is set to zero.

$$\text{GLU}(X) = \sigma(W_4 X + b_3) \odot (W_5 X + b_4). \tag{29}$$

This modular structure enables the GRN to adapt flexibly to different input types and feature combinations, enhancing the Variable Selection Networks' (VSNs) ability to identify and prioritize key variables efficiently.

## C.2 VARIABLE SELECTION NETWORK (VSN)

The variable selection weights $\alpha$ are computed to determine the contribution of each time-variant feature $x_i$ to the aggregated embedding $e^{\text{agg}}$. This is achieved through a Gated Residual Network (GRN) and a softmax function as shown below:

$$\alpha = [\alpha_1, \ldots, \alpha_i, \ldots, \alpha_m] = \text{Softmax}(\text{GRN}([e_1, \ldots, e_i, \ldots, e_m], c_s)), \tag{30}$$

where $c_s$ is the static covariate encoder and $e_i$ is the embedding vector of feature $x_i$. The aggregated entity embedding vector $e^{\text{agg}}$ is a weighted sum of all the $m$ time-variant variable embeddings:

$$e^{\text{agg}} = \sum_{i=1}^{m} \alpha_i \, \text{GRN}(e_i). \tag{31}$$

VSN can be also used for static feature selection, and Figure 7 presents the VSN architecture, with using GRN.

## C.3 TRANSFORMER

The self-attention mechanism in Transformer layers enhances the embeddings by considering the relationships between all elements in the input sequence, allowing the model to capture global context and complex dependencies. The mechanism works by calculating a similarity score between each query ($Q$) and key ($K$) pair, producing attention weights that reflect the importance of each element in relation to others. These attention weights enable each element to be influenced by other relevant elements in the sequence, leading to a dynamic and context-aware representation.

The self-attention mechanism computes the attention weights for a given set of query, key, and value matrices $Q$, $K$, and $V$ as follows:

$$\text{Attention}(Q, K, V) = \text{softmax}\left(\frac{QK^T}{\sqrt{d_k}}\right) V, \tag{32}$$

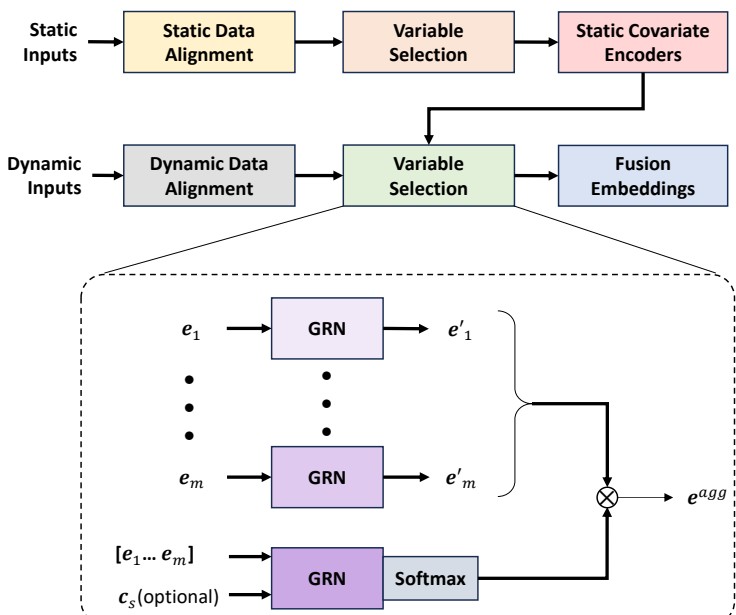

Figure 7: Overview of the entire workflow of the Fusion Layer, where both static and dynamic data pass through two Variable Selection Networks (VSN) with distinct parameters. The static features are selected by themselves, while dynamic data is filtered based on selected static covariates. The calculation mechanism of the VSN is also depicted in the diagram.

where $Q$, $K$, and $V$ are the query, key, and value matrices, respectively. $d_k$ is the dimensionality of the key vectors. The term $\frac{QK^T}{\sqrt{d_k}}$ ensures that the dot-product similarity is normalized by the square root of the dimensionality, preventing large values that could make the softmax function too sharp. The softmax function is applied to the similarity scores to generate a probability distribution, which is then used to weight the values in $V$.

The multi-head attention mechanism allows the model to capture information from multiple representation subspaces. Instead of computing a single attention output, multiple attention heads are computed in parallel, and their results are concatenated and projected back to the original space. The multi-head attention mechanism is defined as:

$$\text{MultiHead}(Q, K, V) = [H_1 \oplus \cdots \oplus H_h \oplus \cdots \oplus H_H]W_H, \tag{33}$$

where $H_h$ represents the output of the $h$-th attention head, computed as:

$$H_h = \text{Attention}(QW_h^Q, KW_h^K, VW_h^V), \tag{34}$$

and $W_h^Q$, $W_h^K$, and $W_h^V$ are learned weight matrices for the query, key, and value matrices, respectively, for the $h$-th head. The symbol $\oplus$ denotes concatenation, meaning the outputs from all attention heads are concatenated into a single vector. $W_H$ is a learned weight matrix that projects the concatenated output back into the model's desired output dimension.

After the multi-head attention step, a feed-forward network (FFN) is applied to introduce non-linearity. The FFN consists of two fully connected layers with a ReLU activation function applied between them. This non-linearity enables the model to capture more complex relationships and dependencies within the data.

Thus, the combination of self-attention and multi-head attention allows the Transformer model to focus on different parts of the input sequence simultaneously, creating a more dynamic and con-

textually aware representation, especially useful for tasks involving long-range dependencies and complex sequence data.

## C.4 Intra-relation Aggregation

To preserve graph heterogeneity and enable fine-grained relation modeling, we perform relation-specific neighborhood aggregation using distinct BiLSTM encoders for each relation type.

At time step $t$, the system is represented as a heterogeneous graph $\mathcal{G}_t = (V, E, R)$, where $R$ denotes the set of edge relation types. For each node $v_i \in V$ and relation $r_\ell \in R$, we aggregate temporal embeddings $h_{j,t}^{\text{temp}}$ from neighbors $v_j \in N_\ell(v_i)$ using:

$$h_{i,\ell}^{\text{agg}}(t) = \frac{1}{|N_\ell(v_i)|} \sum_{v_j \in N_\ell(v_i)} \text{BiLSTM}_\ell(h_{j,t}^{\text{temp}}), \tag{35}$$

Unlike HetGNN, which shares encoders across neighbor types, we assign a distinct BiLSTM per relation type $r_\ell$, allowing the model to disentangle heterogeneous physical or logical interactions. For example, a room might be connected to others via either airflow or control signals—relations that are semantically different and thus require different encoding strategies.

## C.5 Inter-relation Aggregation

To integrate information from multiple relation types, we adopt a multi-head attention mechanism over the aggregated embeddings $h_{i,\ell}^{\text{agg}}(t)$. For each attention head $k = 1, \ldots, K$, attention coefficients $\alpha_\ell^k$ are computed as:

$$\alpha_\ell^k = \text{softmax}\left(\text{LeakyReLU}\left(a^{k\top}[W^k h_{i,t}^{\text{temp}} \| W^k h_{i,\ell}^{\text{agg}}(t)]\right)\right), \tag{36}$$

where $W^k \in \mathbb{R}^{d' \times d}$ is a learnable projection matrix and $a^k \in \mathbb{R}^{2d'}$ is a shared attention vector for the $k$-th head. The final graph-based embedding for node $v_i$ is:

$$h_{i,t}^{\text{graph}} = \frac{1}{K} \sum_{k=1}^{K} \sum_{\ell=1}^{L} \alpha_\ell^k W^k h_{i,\ell}^{\text{agg}}(t), \tag{37}$$

This fusion mechanism allows the model to assign adaptive weights to different relation types per attention head, enabling robust modeling of heterogeneous dependencies. Compared to early fusion approaches, this method provides enhanced flexibility and improved representation quality for nodes participating in multi-relational contexts.

# D Model Version Comparison

This section presents a systematic comparison of different model variants for time-series forecasting in complex building operation systems. All models are trained on the MBS dataset, using the proposed HGTFT architecture. The resulting pretrained model, specialized for the building domain, is termed BOSG (Building Operation System Generator). We explore a range of model configurations by varying embedding dimensions, network depth, and overall parameter count to analyze trade-offs between predictive performance, model size, and training efficiency.

## D.1 Embedding Dimension Adjustment

We tested four embedding dimensions (64, 128, 256, and 512), while keeping the architecture constant: one temporal layer, one graph layer, and two additional temporal layers. Results in Table 6 show that 256-d offers a strong trade-off between accuracy and efficiency. Although 512-d provides marginal MSE improvements, the parameter increase is substantial, with limited performance benefit.

Table 6: Performance comparison of various model configurations with different embedding dimensions.

| Embedding dimension | Model size | MSE | RCS | CRS | FDS |
|---|---|---|---|---|---|
| 64-d | 22,241,773 | 0.0098 | 0.0168 | 0.434 | 0.491 |
| 128-d | 81,437,154 | 0.0059 | 0.0045 | 0.396 | 0.448 |
| 256-d | 310,800,689 | 0.0027 | **0.0012** | **0.312** | **0.405** |
| 512-d | 1,173,418,849 | **0.0026** | **0.0012** | 0.320 | 0.413 |

## D.2 MODEL LAYER ADJUSTMENT

We compared multiple network layer configurations, modifying the order and count of temporal and graph layers (see Table 7). Results indicate that placing a temporal layer before the graph layer is essential for capturing temporal context prior to modeling inter-object relations. Additional temporal layers after the graph layer further improve performance, but benefits plateau beyond two layers.

Table 7: Performance comparison of various model configurations with different network layer architectures.

| Layer configuration | Model size | MSE | RCS | CRS | FDS |
|---|---|---|---|---|---|
| Graph+Temporal | 197,075,249 | 0.0068 | 0.0025 | 0.487 | 0.536 |
| Temporal+Graph | 197,075,249 | 0.0056 | 0.0021 | 0.469 | 0.480 |
| Temporal+Graph×2 | 213,697,841 | 0.0053 | 0.0020 | 0.454 | 0.477 |
| (Temporal+Graph)×2 | 270,560,561 | 0.0039 | 0.0018 | 0.413 | 0.439 |
| (Temporal+Graph)×3 | 344,045,873 | 0.0033 | 0.0015 | 0.375 | 0.414 |
| Temporal+Graph+Temporal | 253,937,969 | 0.0037 | 0.0016 | 0.395 | 0.468 |
| Temporal×2+Graph+Temporal | 310,800,689 | 0.0036 | 0.0016 | 0.386 | 0.442 |
| Temporal×2+Graph+Temporal×2 | 367,663,409 | 0.0028 | 0.0013 | 0.306 | 0.399 |
| Temporal+Graph+Temporal×2 | 310,800,689 | 0.0027 | **0.0012** | 0.312 | 0.405 |
| Temporal+Graph+Temporal×3 | 367,663,409 | **0.0026** | **0.0012** | **0.304** | **0.397** |

## D.3 SCALING STUDY AND MODEL VARIANTS

We conducted a scaling study on the BOSG model to investigate the relationship between model size, computation, and forecasting performance. Four BOSG configurations were trained with parameter sizes of 20M, 80M, 310M, and 1.26B, each using 30K iterations and a fixed global batch size of 64. All model variants adopted 8 attention heads and incorporated up/down projection layers to enhance feature representation. Their architectural details and evaluation results are summarized in Table 8. As model size increased, the primary forecasting metric (MSE) consistently decreased from 0.0107 to 0.0025, with notable gains up to 310M parameters. However, performance improvement between the 310M and 1.26B models was marginal, indicating diminishing returns at larger scales.

To better understand compute-performance efficiency, we saved model checkpoints at specific FLOPS intervals during training and plotted the resulting MSE values on a log scale. As shown in Figure 8, training performance improved with increasing computational budget, although the rate of improvement flattened beyond the 310M model. All experiments were conducted on a high-performance system consisting of eight NVIDIA A800 GPUs (80GB memory each), providing 3456 tensor cores in total. This setup enabled efficient parallel training, with the largest model (1.26B) completing 30K iterations in approximately three days. These findings provide practical guidance for compute-optimal scaling in time-series modeling.

Table 8: Performance comparison of BOSG model variants with varying parameter sizes and configurations.

| Version | Params | Embedding | Layer configuration | MSE | RCS | CRS | FDS |
|---------|--------|-----------|--------------------|-----|-----|-----|-----|
| 20M | 19,164,316 | 64-d | Graph+Temporal | 0.0107 | 0.0184 | 0.496 | 0.519 |
| 80M | 77,202,922 | 128-d | Temporal+Graph+Temporal | 0.0073 | 0.0055 | 0.435 | 0.481 |
| 310M | 310,800,689 | 256-d | Temporal+Graph+Temporal×2 | 0.0027 | **0.0012** | 0.312 | **0.405** |
| 1.26B | 1,258,271,153 | 512-d | Temporal+Graph+Temporal×3 | **0.0025** | **0.0012** | **0.307** | 0.416 |

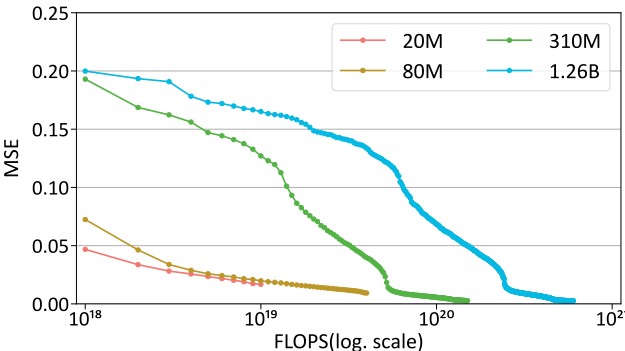

Figure 8: Training MSE vs. FLOPS (log scale) for different BOSG model sizes (20M, 80M, 310M, 1.26B) on the SSL masked time-series modeling task.

# E  SELF-SUPERVISED LEARNING

## E.1  LOSS FUNCTION DETAILS FOR SELF-SUPERVISED LEARNING

The following describes the loss functions employed in our self-supervised learning (SSL) tasks, aimed at ensuring clarity and reproducibility. For masked time-series modeling, the reconstruction error is quantified using the Mean Squared Error (MSE) as follows:

$$\text{MSE} = \frac{1}{N} \sum_{i=1}^{N} \left( \frac{1}{F_i} \sum_{f=1}^{F_i} \left( \frac{1}{M_{i,f}} \sum_{t=1}^{M_{i,f}} (y_{i,f}(t) - \hat{y}_{i,f}(t))^2 \right) \right), \tag{38}$$

where $N$ represents the number of nodes, $F_i$ denotes the number of features for the $i$-th node, and $M_{i,f}$ is the number of masked time points for each feature.

For the graph-based task, the Binary Cross-Entropy (BCE) loss function is utilized to evaluate the classification accuracy of edge predictions, defined as:

$$\text{BCE} = -\frac{1}{N_r} \sum_{i=1}^{N_r} (r_i \log(\hat{r}_i) + (1 - r_i) \log(1 - \hat{r}_i)), \tag{39}$$

where $N_r$ represents the number of samples, $\hat{r}_i$ is the predicted relation, and $r_i$ denotes the true relation value.

## E.2  MODEL TRAINING EXPERIMENTS FOR SELF-SUPERVISED RELATIONSHIP LEARNING TASK

We conducted experiments for self-supervised relationship learning task with the HGTFT model to identify which network layers are essential to update and which can remain fixed. Additionally, we evaluated the prediction results when the parameters of the task-specific linear transformation layer were either initialized randomly without updates or jointly updated alongside HGTFT. Further, we examined the effect of initializing HGTFT parameters either randomly or using pre-trained

weights from a masked time-series modeling task. The results of these validation experiments are summarized in Table 9.

Table 9: Experimental results of masked edge modeling for various model update approaches.

| Case No. | HGTFT update layer | Task NW | Initialization | loss (BCE) |
|---|---|---|---|---|
| Case 1 | node, temporal, graph | Update | Random | 0.34 |
| Case 2 | temporal, graph | Update | Random | 0.35 |
| Case 3 | graph | Update | Random | 0.42 |
| Case 4 | temporal, graph | w/o update | Random | 0.35 |
| Case 5 | temporal, graph | w/o update | Masked time-series modeling | **0.28** |

The experimental results revealed that updating the network layers responsible for the temporal and graph embeddings is crucial for task performance. Additionally, reusing the pre-trained parameters from the masked time-series modeling task provided a significant improvement over random initialization. Interestingly, the task-specific linear output layer primarily acted as a dimensionality transformation and had minimal impact on the prediction results. Based on these observations, we determined that the optimal approach involves initializing the base HGTFT model parameters from the trained masked time-series modeling task, updating only the temporal and graph embeddings, and leaving the task-specific linear output layer randomly initialized and fixed during training.

### E.3 Training Pipeline for Self-Supervised Learning

In our self-supervised learning approach, we prioritized the masked time-series modeling task as the primary objective, with the masked edge modeling task as a secondary target. The goal was to minimize the loss of the masked edge modeling task while ensuring that the loss of the masked time-series modeling task increased by no more than 10% from its optimal value. A series of sequential training experiments were conducted to achieve this balance, and the results are summarized in Table 10.

Table 10: Experiment results for the self-supervised learning pipeline.

| Step No. | Masked time-series modeling | | | Masked edge modeling | | |
|---|---|---|---|---|---|---|
| | Task on/off | Starting loss | Ending loss | Task on/off | Starting loss | Ending loss |
| Step 1 | On | 1.8421 | 0.0027 | Off | N/A | 0.6942 |
| Step 2 | Off | 0.0027 | 0.6439 | On | 0.6942 | 0.2885 |
| Step 3 | On | 0.6439 | 0.0028 | Off | 0.2885 | 0.4526 |
| Step 4 | Off | 0.0028 | 0.2781 | On | 0.4526 | 0.2640 |
| Step 5 | On | 0.2781 | 0.0026 | Off | 0.2640 | 0.3304 |
| Step 6 | Off | 0.0026 | 0.2673 | On | 0.3304 | 0.2595 |
| Step 7 | On | 0.2673 | 0.0026 | Off | 0.2595 | 0.3184 |

Through a total of seven rounds of alternating training between the two tasks, we observe a consistent decrease in the loss for the masked time-series modeling task before each training session, with little change in the loss after training. In contrast, for the masked edge modeling task, the loss values showed noticeable reductions both before and after training in each round. Notably, the final round of training for the masked time-series modeling task had minimal impact on the graph relationship prediction, suggesting that the model had converged and further training on this task no longer significantly affected the performance of the masked edge modeling task.

## F Supervised Learning

### F.1 Supervised Learning subtask model

Each subtask shares a unified decoder structure (see Figure 9), where masked attention connects historical embeddings to future targets. GRN blocks and lightweight dense projections are included

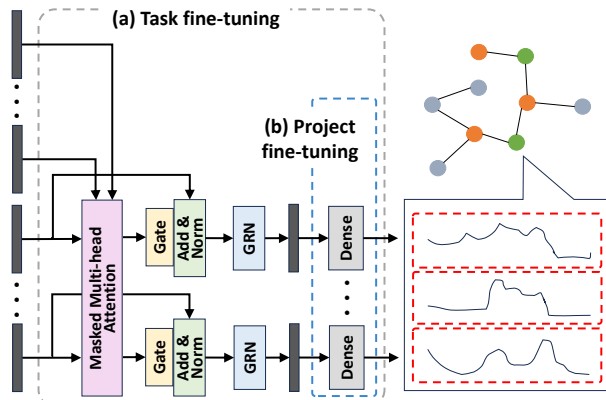

Figure 9: Model structure for a typical prediction subtask with two fine-tuning phases.

for stable adaptation. Fine-tuning is performed in two stages: task-level tuning updates only task-specific parameters, while project-level tuning adjusts the dense head to align with limited real-world data, preserving general representations learned during pretraining.

### F.2    SUPERVISED LEARNING TRAINING TASK

Forecasting tasks in multiphysics systems exhibit substantial diversity due to the heterogeneity of entities, variable types, and interaction structures. To capture this complexity, we construct a suite of supervised learning tasks based on scenario-specific interaction topologies. Each scenario is represented as a heterogeneous graph comprising distinct physical entities (e.g., thermal zones, fluid circulation units, environmental sensors) and their relationships, as illustrated in Figure 10 for scenario 3.3.

Beyond structural diversity, variations in variable availability across entities further contribute to task differentiation. We first define original tasks by selecting strongly correlated entities and predicting all of their dynamic variables for future time points. Derived tasks are then generated by selectively masking or revealing subsets of variables in the future, simulating diverse observability conditions. An example of such task construction is provided in Table 5.

### F.3    MEAN SQUARE ERROR FOR SUPERVISED LEARNING

The accuracy loss, denoted as $L_{\mathrm{MSE}}$, is quantified using the Mean Squared Error (MSE) across all entities for each task, as formally defined below:

$$L_{\mathrm{MSE}} = \frac{1}{N} \sum_{i=1}^{N} \left( y_i(t, T_{\mathrm{future}}) - \hat{y}_i(t, T_{\mathrm{future}}) \right)^2 , \tag{40}$$

where $N$ represents the number of entities, which may vary across different tasks. The terms $y_i(t, T_{\mathrm{future}})$ and $\hat{y}_i(t, T_{\mathrm{future}})$ refer to the true and predicted values, respectively, for the time period from $t + 1$ to $t + T_{\mathrm{future}}$, corresponding to all dynamic prediction features of the $i$-th entity. For the sake of brevity and clarity, the feature dimension is omitted from the formula.

### F.4    REASONABLENESS CHECKS SCORE

In complex physical systems, time series predictions must not only achieve numerical accuracy but also respect fundamental physical laws and operational constraints. We propose the *Reasonableness Checks Score* (RCS) as an auxiliary evaluation metric to quantify the degree to which predicted values conform to domain-specific physical expectations. Rather than being limited to any particular field, the RCS framework is designed to be modular and extensible, supporting multi-domain constraints across various physical and engineered systems.

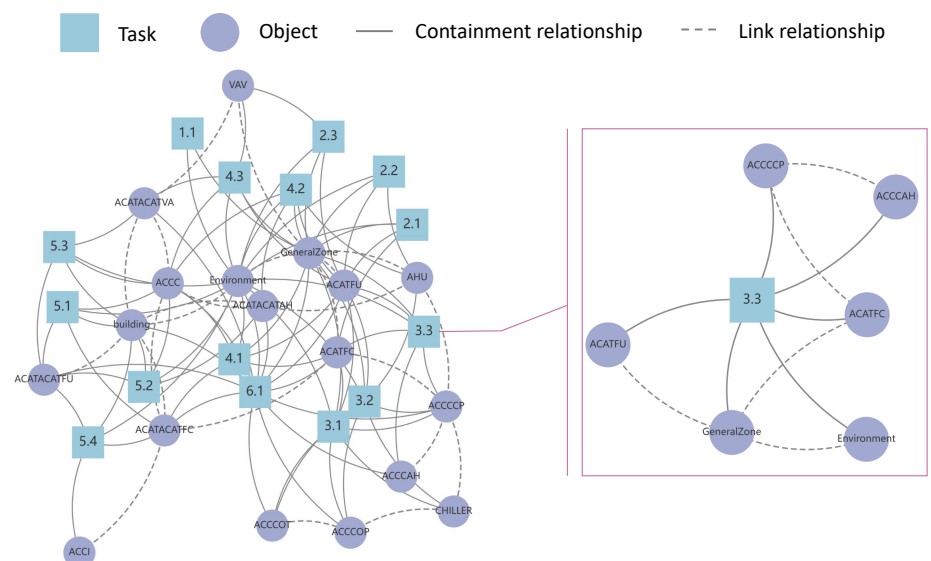

Figure 10: Topology of tasks and entity types in a building multiphysics system. Each scenario defines a specific combination of interconnected entities, identified by a unique scenario ID. Highlighted example 3.3 includes Environment, General Zone, Chiller, Chilled/ Cooling Water Pumps (ACCCCP/ACCCOP), Cooling Tower (ACCCOT), Fan Coil Unit (ACATFC), and Supply Air Fan (ACATFU).

To structure this assessment, we categorize reasonableness checks into four generalized dimensions:

1. **Physical State Bounds:** Core physical quantities (e.g., temperature, pressure, flow rate, power) should remain within known feasible or safe ranges, derived from empirical knowledge or physical laws.

2. **Energy and Resource Balance:** Energy usage, mass flow, or material consumption should be consistent with input-output relationships and operational schedules. Sudden discontinuities or unrealistic surges may indicate violations of conservation principles or faulty control.

3. **System Operating Constraints:** Devices or subsystems should operate in valid configurations, respecting timing constraints, activation conditions, and logical dependencies (e.g., cooling should not activate when the system is already below the lower threshold).

4. **Inter-Component Consistency:** Multiple subsystems interacting within the same environment should exhibit consistent behavior. For example, responses to a shared external stimulus should not contradict each other or physical causality.

Each reasonableness check can be modeled as a differentiable function that penalizes violations of soft physical constraints. The total RCS loss is defined as:

$$L_{\mathrm{RCS}} = \sum_{k=1}^{K} g_k\left(\hat{y}_k(t, T_{\mathrm{future}})\right), \tag{41}$$

where $g_k(\cdot)$ denotes the $k$-th check function applied to the predicted output $\hat{y}_k(t, T_{\mathrm{future}})$, and $K$ is the total number of checks relevant to the task.

**Example 1: Bounded Range Check.** For a physical variable $\hat{y}(t)$ constrained within a range $[y_{\min}, y_{\max}]$, the penalty term can be formulated as:

$$g_{\mathrm{range}}(\hat{y}(t)) = \lambda \cdot \left[\max\left(0, \hat{y}(t) - y_{\max}\right)^2 + \max\left(0, y_{\min} - \hat{y}(t)\right)^2\right], \tag{42}$$

where $\lambda$ is a weighting coefficient controlling the penalty strength at each time point $t$.

**Example 2: Energy Conservation Check.** In multi-physical systems, the principle of energy conservation often serves as a key constraint. For instance, in a thermal process involving heat exchange, the heat entering a system at time $t$ should approximately equal the sum of the heat leaving the system and the internal losses, i.e., $Q_{\text{in}}(t) \approx Q_{\text{out}}(t) + Q_{\text{loss}}(t)$. To enforce this physical constraint on predicted outputs, we define the energy conservation check function as:

$$g_{\text{energy}}(t) = \gamma \cdot \left( \hat{Q}_{\text{in}}(t) - \hat{Q}_{\text{out}}(t) - Q_{\text{loss}}(t) \right)^2, \tag{43}$$

where $\hat{Q}_{\text{in}}(t)$ and $\hat{Q}_{\text{out}}(t)$ denote the predicted input and output energy at time $t$, and $Q_{\text{loss}}(t)$ is a predefined (or estimated) time-dependent loss term. The scalar $\gamma$ controls the importance of this check. This function penalizes deviations from the expected energy balance at each timestep, thereby promoting physically consistent predictions.

### F.5 CORRELATION-BASED SCORE

The Correlation-Based Score (CBS) evaluates the statistical correlation between predicted and true values in time-series forecasting by computing the Pearson correlation coefficients for both predicted and true values, determining the deviation between these correlations for each variable pair, and then calculating the loss as the Mean Squared Error (MSE) of these deviations.

The formula for the CBS loss $L_{\text{corr}}$ is given by:

$$L_{\text{CBS}} = \frac{1}{L} \sum_{l=1}^{L} \left( |\rho(\hat{y}_i, \hat{y}_j) - \rho(y_i, y_j)|^2 \right), \tag{44}$$

Where $L$ is the number of variable pairs in the prediction task. $\rho(\hat{y}_i, \hat{y}_j)$ is the Pearson correlation coefficient between the predicted values $\hat{y}_i$ and $\hat{y}_j$, and $\rho(y_i, y_j)$ is the Pearson correlation coefficient between the true values $y_i$ and $y_j$.

### F.6 FREQUENCY DOMAIN SIMILARITY

To calculate the similarity between two time-series datasets in the frequency domain, we can use the Fourier Transform to convert both datasets from the time domain to the frequency domain and then compare their frequency components. The steps are stated as following:

1. **Fourier Transform:** Apply the Fourier Transform to each time series to obtain the amplitude and phase spectra. Let $A_X(f)$ and $\theta_X(f)$ be the amplitude and phase of the first time-series data across frequencies $f$. Similarly, $A_Y(f)$ and $\theta_Y(f)$ represent the amplitude and phase of the second time series.

2. **Amplitude Cosine Similarity:** Define the cosine similarity for the amplitude spectra of the two time-series datasets as follows:

$$S_{\text{amp}} = \frac{\sum_{f=1}^{N} A_X(f) \cdot A_Y(f)}{\sqrt{\sum_{f=1}^{N} A_X(f)^2} \cdot \sqrt{\sum_{f=1}^{N} A_Y(f)^2}} \tag{45}$$

   where $N$ is the number of frequency components. This metric evaluates the similarity in amplitude between the two datasets.

3. **Phase Cosine Similarity:** Define the cosine similarity for the phase spectra by converting the phase angles into their respective sine and cosine components:

$$S_{\text{phase}} = \frac{\sum_{f=1}^{N} \left( \cos(\theta_X(f)) \cdot \cos(\theta_Y(f)) + \sin(\theta_X(f)) \cdot \sin(\theta_Y(f)) \right)}{\sqrt{\sum_{f=1}^{N} \left( \cos(\theta_X(f))^2 + \sin(\theta_X(f))^2 \right)} \cdot \sqrt{\sum_{f=1}^{N} \left( \cos(\theta_Y(f))^2 + \sin(\theta_Y(f))^2 \right)}} \tag{46}$$

   This metric evaluates the alignment of phase angles between the two time series.

Table 11: Supervised learning pipeline with loss weights.

| Stage No. | Foundation model update | Task model update | Learning rate | $a_1$ | $a_2$ | $a_3$ | $a_4$ |
|-----------|------------------------|-------------------|---------------|-------|-------|-------|-------|
| Stage 1 | N | Y | 0.1 | 1 | 0 | 0 | 0 |
| Stage 2 | Y | Y | 0.01 | 0.5 | 0.1 | 0.2 | 0.2 |
| Stage 3 | Y | N | 0.005 | 0.7 | 0.1 | 0.1 | 0.1 |
| Stage 4 | Y | N | 0.003 | 0.7 | 0.3 | 0 | 0 |
| Stage 5 | Y | N | 0.001 | 0.5 | 0.5 | 0 | 0 |

4. **Combined Frequency Domain Similarity:** Finally, define the combined frequency domain similarity $S_{\text{freq}}$ using a weighted sum of the amplitude and phase similarities:

$$S_{\text{freq}} = \alpha S_{\text{amp}} + \beta S_{\text{phase}} \tag{47}$$

where $\alpha$ and $\beta$ are weights that can be adjusted based on the relative importance of amplitude and phase similarity. This combined metric $S_{\text{freq}}$ captures both amplitude and phase alignment, offering a comprehensive measure of similarity between the two time-series datasets in the frequency domain. The loss for Frequency Domain Similarity (FDS), $L_{\text{FDS}}$, is $1 - S_{\text{freq}}$.

### F.7 SUPERVISED LEARNING PIPELINE

The supervised learning pipeline is organized into five stages to enhance the foundation model's ability to aggregate and represent information, building on the self-supervised phase and improving its applicability to generalizable time-series prediction tasks. The primary objective is to enhance the adaptability and representational capacity of the foundation model, rather than focusing solely on maximizing accuracy for individual time-series subtasks. Each stage refines a specific aspect of the model, as summarized in Table 11. The total loss for task $i$ is computed as

$$L_{\text{task},i} = a_1 L_{\text{MSE}} + a_2 L_{\text{RCS}} + a_3 L_{\text{CRS}} + a_4 L_{\text{FDS}}, \tag{48}$$

where $a_1, a_2, a_3, a_4$ denote the respective weights of each loss component.

In Stage 1, the parameters of the foundation model, initialized from the self-supervised phase, are frozen, with only the task-specific parameters being updated. This allows the model to quickly adapt to a reasonable accuracy range, using a learning rate of 0.1, while prioritizing the MSE loss function.

In Stage 2, both the foundation model and task-specific models are jointly trained, with the learning rate gradually decaying from 0.1 to 0.01. This stage aims to improve prediction accuracy and gradually bring it closer to optimal performance. The loss weights are adjusted to strike a balanced consideration of the different loss functions.

In Stages 3, 4, and 5, the task-specific parameters are frozen, and the foundation model is further refined to enhance generalization capability. The learning rate is progressively reduced to 0.005, 0.003, and 0.001, respectively. During these stages, the loss weights are adjusted to refine model performance. In Stage 3, the focus is on improving accuracy with minimal adjustments to the consistency and rationality losses. In Stages 4 and 5, the loss weights are updated to place greater emphasis on $L_{\text{RCS}}$, promoting improved rationality and domain-specific reasoning.

Data is allocated across the five stages following a 1:3:4:1:1 ratio. The majority of the data is utilized during the second and third stages for joint training and generalization, while the final stages concentrate on fine-tuning the foundation model for improved rationality and consistency.

## G BASELINE METHODS SELECTION

To evaluate the performance of HGTFT across zero-shot and few-shot forecasting tasks, we compare it against diverse baselines, including classic models (No LMs), time-series large models (Time LMs), and large language model-based methods (LLMs). These methods differ in how they handle input modalities such as time-series (TS), static metadata (Static), and graph structure (Graph), as

detailed in Table 12. For each method, we selected the most capable open-source version available to ensure a fair comparison.

Below, we provide an overview of the selected baseline methods and their respective adaptations to our setting:

- **LSTM** and **Autoformer**: Forecast each variable independently, without incorporating static or relational information. Their outputs are aggregated through post-processing to construct full multivariate predictions.

- **TFT**: Combines multivariate time-series data with static features to perform object-level forecasting. It supports variable selection and interpretable attention mechanisms but does not model inter-object dependencies.

- **HTGNN**: Utilizes graph-structured time-series inputs, leveraging the relationships between objects to perform dynamic variable forecasting in a heterogeneous setting.

- **STD-MAE**: Utilizes graph-structured time-series inputs in a homogeneous setting, where the system is decomposed into multiple homogeneous subgraphs. Each subgraph is modeled independently to capture localized spatial-temporal patterns, and the predictions are subsequently aggregated to form the overall system-level forecast.

- **TimesFM** and **MOIRAI**: Encode each object type's time-series data sequentially, forecasting each variable independently. These models do not utilize static or graph information; instead, multivariate predictions are obtained by batching univariate forecasts.

- **LLMTime** and **Time-LLM**: Process multiple object instances simultaneously using only time-series data. These LLM-based models do not account for static metadata or inter-instance relationships, but benefit from large-scale pretraining and context-aware generation.

Table 12: Baseline methods summary.

| Method | Input Type | Category | Model Version |
|---|---|---|---|
| LSTM | TS | No LM | — |
| Autoformer | TS | No LM | — |
| TFT | TS, Static | No LM | — |
| HTGNN | TS, Graph | No LM | — |
| STD-MAE | TS, Graph | No LM | — |
| TimesFM | TS | Time LM | 200M |
| MOIRAI | TS | Time LM | 1.1-R-large |
| LLMTime | TS | LLM | LLaMA-2 70B |
| Time-LLM | TS | LLM | LLaMA 7B |

In recent years, there has been a large number of work focusing on spatial-temporal forecasting in relatively simple settings involving homogeneous object types and graph structures. Although these works differ from the problem definition and setting in our study, we include several widely recognized spatial-temporal forecasting algorithms from the past 4 years for a more comprehensive comparison. We evaluate their performance on four standard datasets: PEMS04, PEMS08, COVID-19 (JHU), and COVID-19 (NYT). As shown in Table 13, while recent methods continue to make marginal improvements in these benchmarks, the performance gap is narrowing. This highlights a critical limitation: the lack of methods and datasets capable of handling more complex scenarios. Addressing this gap is the primary motivation of our work.

We also evaluate our method and selected baselines on commonly used standard time-series datasets, including ETT, Weather, Electricity, Traffic and ILI. Although these datasets are primarily benchmarks for purely data-driven forecasting and are not the main focus of our study, our method achieves performance comparable to state-of-the-art models (Table 14).

## H  NORMALIZATION METHODS

To evaluate the effectiveness of different normalization methods, using MSE directly on normalized data is not appropriate, as each method applies a unique scaling to the variables, which would

Table 13: Performance comparison on PEMS04, PEMS08, COVID-19 (JHU), COVID-19 (NYT) datasets. Best results are in **bold**, second best are underlined.

| Model | PEMS04 | | PEMS08 | | COVID-19 (JHU) | | COVID-19 (NYT) | |
|---|---|---|---|---|---|---|---|---|
| | MAE | RMSE | MAE | RMSE | MAE | RMSE | MAE | RMSE |
| LSTM Hochreiter (1997) | $32.48 \pm 0.38$ | $55.51 \pm 0.74$ | $24.98 \pm 0.38$ | $41.71 \pm 0.43$ | $122.42 \pm 1.41$ | $232.11 \pm 3.51$ | $70.59 \pm 0.85$ | $139.18 \pm 1.71$ |
| Autoformer Wu et al. (2021) | $32.39 \pm 0.43$ | $53.19 \pm 0.75$ | $25.56 \pm 0.34$ | $41.65 \pm 0.44$ | $115.77 \pm 1.08$ | $198.67 \pm 2.48$ | $62.37 \pm 0.76$ | $133.35 \pm 1.68$ |
| TFT Lim et al. (2021) | $31.32 \pm 0.35$ | $48.37 \pm 0.67$ | $24.63 \pm 0.36$ | $39.74 \pm 0.42$ | $121.81 \pm 1.43$ | $261.77 \pm 3.20$ | $71.36 \pm 0.84$ | $158.93 \pm 2.30$ |
| STformer Grigsby et al. (2021) | $31.69 \pm 0.48$ | $55.70 \pm 0.71$ | $24.91 \pm 0.44$ | $43.23 \pm 0.59$ | $72.42 \pm 0.90$ | $166.86 \pm 2.13$ | $49.63 \pm 0.64$ | $123.01 \pm 1.51$ |
| TimesFM Das et al. (2023) | $32.57 \pm 0.43$ | $55.94 \pm 0.68$ | $23.93 \pm 0.38$ | $42.41 \pm 0.56$ | $99.75 \pm 1.10$ | $216.63 \pm 2.99$ | $57.07 \pm 0.72$ | $113.45 \pm 1.62$ |
| MOIRAI Woo et al. (2024) | $33.31 \pm 0.45$ | $55.51 \pm 0.72$ | $24.03 \pm 0.32$ | $42.49 \pm 0.56$ | $105.74 \pm 1.26$ | $234.24 \pm 2.70$ | $81.05 \pm 0.96$ | $134.57 \pm 1.94$ |
| LLMTime Gruver et al. (2024) | $33.69 \pm 0.52$ | $52.49 \pm 0.76$ | $26.68 \pm 0.40$ | $43.94 \pm 0.44$ | $115.37 \pm 1.22$ | $216.74 \pm 2.71$ | $72.24 \pm 0.90$ | $157.31 \pm 1.87$ |
| Time-LLM Jin et al. (2023) | $32.23 \pm 0.40$ | $52.18 \pm 0.67$ | $27.74 \pm 0.48$ | $40.01 \pm 0.44$ | $95.01 \pm 1.13$ | $201.14 \pm 2.68$ | $83.17 \pm 1.01$ | $146.45 \pm 2.02$ |
| ASTGCN Guo et al. (2019) | $23.46 \pm 0.27$ | $34.88 \pm 0.59$ | $17.91 \pm 0.31$ | $28.80 \pm 0.47$ | $58.10 \pm 0.61$ | $109.14 \pm 1.91$ | $33.71 \pm 0.48$ | $93.55 \pm 1.37$ |
| STGCN Han et al. (2020) | $21.72 \pm 0.34$ | $34.61 \pm 0.57$ | $18.73 \pm 0.35$ | $28.05 \pm 0.50$ | $52.94 \pm 0.63$ | $110.63 \pm 1.68$ | $37.25 \pm 0.44$ | $88.62 \pm 1.27$ |
| STSGCN Song et al. (2020) | $21.26 \pm 0.28$ | $34.42 \pm 0.50$ | $17.86 \pm 0.32$ | $27.45 \pm 0.45$ | $53.19 \pm 0.55$ | $111.51 \pm 1.65$ | $34.58 \pm 0.46$ | $89.42 \pm 1.31$ |
| STFGNN Li & Zhu (2021) | $19.24 \pm 0.25$ | $31.18 \pm 0.55$ | $16.76 \pm 0.29$ | $25.74 \pm 0.44$ | $51.45 \pm 0.56$ | $101.48 \pm 1.72$ | $33.00 \pm 0.48$ | $82.17 \pm 1.38$ |
| STGODE Fang et al. (2021) | $21.63 \pm 0.31$ | $33.30 \pm 0.49$ | $16.14 \pm 0.31$ | $25.46 \pm 0.45$ | $56.84 \pm 0.66$ | $106.01 \pm 1.66$ | $32.09 \pm 0.44$ | $81.72 \pm 1.33$ |
| STNorm Deng et al. (2021) | $19.07 \pm 0.28$ | $31.91 \pm 0.52$ | $15.05 \pm 0.27$ | $25.64 \pm 0.41$ | $46.68 \pm 0.53$ | $99.34 \pm 1.58$ | $30.59 \pm 0.72$ | $80.32 \pm 1.21$ |
| DSTAGNN Lan et al. (2022) | $19.87 \pm 0.33$ | $30.80 \pm 0.53$ | $15.95 \pm 0.32$ | $24.53 \pm 0.38$ | $50.45 \pm 0.57$ | $100.51 \pm 1.57$ | $31.49 \pm 0.46$ | $76.50 \pm 1.31$ |
| HTGNN Fan et al. (2022) | $21.01 \pm 0.37$ | $36.44 \pm 0.56$ | $18.22 \pm 0.38$ | $27.04 \pm 0.48$ | $46.24 \pm 0.51$ | $102.73 \pm 1.56$ | $31.16 \pm 0.49$ | $75.98 \pm 1.29$ |
| PDFormer Jiang et al. (2023) | $18.60 \pm 0.29$ | $29.94 \pm 0.51$ | **$12.82 \pm 0.26$** | $22.62 \pm 0.35$ | $46.57 \pm 0.58$ | $93.48 \pm 1.21$ | $28.69 \pm 0.44$ | $71.70 \pm 1.12$ |
| STAEformer Liu et al. (2023a) | $18.62 \pm 0.30$ | **$29.65 \pm 0.44$** | $12.97 \pm 0.26$ | $24.21 \pm 0.35$ | $47.58 \pm 0.59$ | $96.87 \pm 1.34$ | $24.62 \pm 0.41$ | $77.43 \pm 1.30$ |
| STD-MAE Gao et al. (2023) | **$17.85 \pm 0.27$** | $29.72 \pm 0.44$ | $13.67 \pm 0.28$ | **$22.62 \pm 0.36$** | $47.75 \pm 0.60$ | **$92.62 \pm 1.27$** | $26.69 \pm 0.45$ | $72.98 \pm 1.21$ |
| **HGTFT (Ours)** | $19.94 \pm 0.34$ | $32.16 \pm 0.54$ | $16.43 \pm 0.34$ | $25.08 \pm 0.41$ | **$41.54 \pm 0.44$** | $94.38 \pm 1.20$ | $25.69 \pm 0.42$ | **$65.64 \pm 1.04$** |

Table 14: Performance on standard time-series forecasting. ETT results are averaged over four subsets: ETTh1, ETTh2, ETTm1, and ETTm2. All models are trained or fine-tuned on 10% of each dataset. Best results are in **bold**, second best are underlined.

| Dataset | Metric | LSTM | Autoformer | TFT | HTGNN | STD-MAE | TimesFM | MOIRAI | LLMTime | Time-LLM | HGTFT (Ours) |
|---|---|---|---|---|---|---|---|---|---|---|---|
| ETT | MSE | 0.589 | 0.465 | 0.400 | 0.455 | 0.480 | 0.421 | **0.391** | 0.575 | 0.408 | 0.425 |
| | MAE | 0.597 | 0.459 | 0.412 | 0.484 | 0.534 | 0.437 | **0.404** | 0.577 | 0.428 | 0.441 |
| Weather | MSE | 0.332 | 0.338 | 0.292 | 0.335 | 0.393 | 0.299 | 0.259 | 0.345 | **0.237** | 0.299 |
| | MAE | 0.363 | 0.382 | 0.311 | 0.366 | 0.383 | 0.321 | 0.287 | 0.412 | **0.264** | 0.334 |
| Electricity | MSE | 0.268 | 0.227 | 0.239 | 0.263 | 0.257 | 0.245 | 0.192 | 0.276 | **0.163** | 0.219 |
| | MAE | 0.365 | 0.338 | 0.318 | 0.358 | 0.394 | 0.330 | 0.295 | 0.390 | **0.264** | 0.317 |
| Traffic | MSE | 0.804 | 0.628 | 0.646 | 0.552 | 0.596 | 0.521 | 0.620 | 0.813 | **0.383** | 0.481 |
| | MAE | 0.509 | 0.379 | 0.398 | 0.389 | 0.433 | 0.344 | 0.336 | 0.498 | **0.264** | 0.350 |
| ILI | MSE | 4.753 | 3.125 | 3.343 | 4.365 | 3.894 | 2.435 | 1.573 | 2.868 | **1.437** | 2.432 |
| | MAE | 1.580 | 1.168 | 1.281 | 1.550 | 1.489 | 1.021 | 0.935 | 1.047 | **0.805** | 1.077 |

make MSE comparisons unfair. Instead, we reverse-normalize the variables before calculating the evaluation metrics to ensure a fair comparison of methods. However, the diverse ranges of the original variables after reverse normalization pose challenges in balancing weights across variables. To address this, we focus on key variables from the training tasks and compute statistical metrics for each individually. Their CV-RMSE values are listed in Table 15. As shown, the "Multi-Instance Normalization" method achieves more balanced prediction performance across various variables compared to the other methods.

## I TIME SERIES FORECASTING VISUALIZATION

To facilitate a qualitative analysis of the zero-shot and few-shot prediction results based on the BOSG-310M, we present time-series prediction plots for several key variables. The plots illustrate the forecasting performance of the proposed model on three critical objects: room, fan coil unit (FCU), and chiller system, under both zero-shot and fine-tuned conditions.

As shown in Figures 11 to 13, the zero-shot predictions capture the overall trends and patterns for each variable, although the accuracy of the predictions varies across different variables. While the model is able to predict the general shape of the curves, the degree of precision differs, reflecting the inherent challenges of making predictions without prior task-specific fine-tuning. In addition, we present the results of predictions following fine-tuning with one month of data. The improvements are evident, with significantly enhanced accuracy across all variables, particularly in capturing short-term dynamics. However, it is important to note that fine-tuning with a relatively short period of data, although it improves predictions for recent time period and conditions similar to those seen

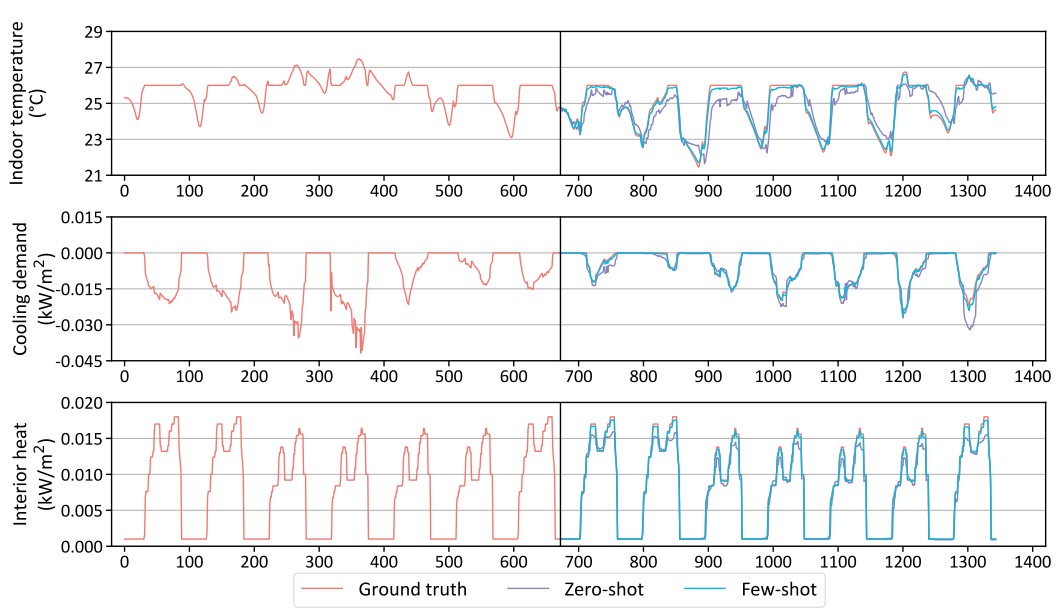

Figure 11: Visualization of time-series forecasting for key variables of a room object, predicting the next 7 days based on the past 7 days. Predictions include zero-shot and few-shot (with one month of fine-tuning data).

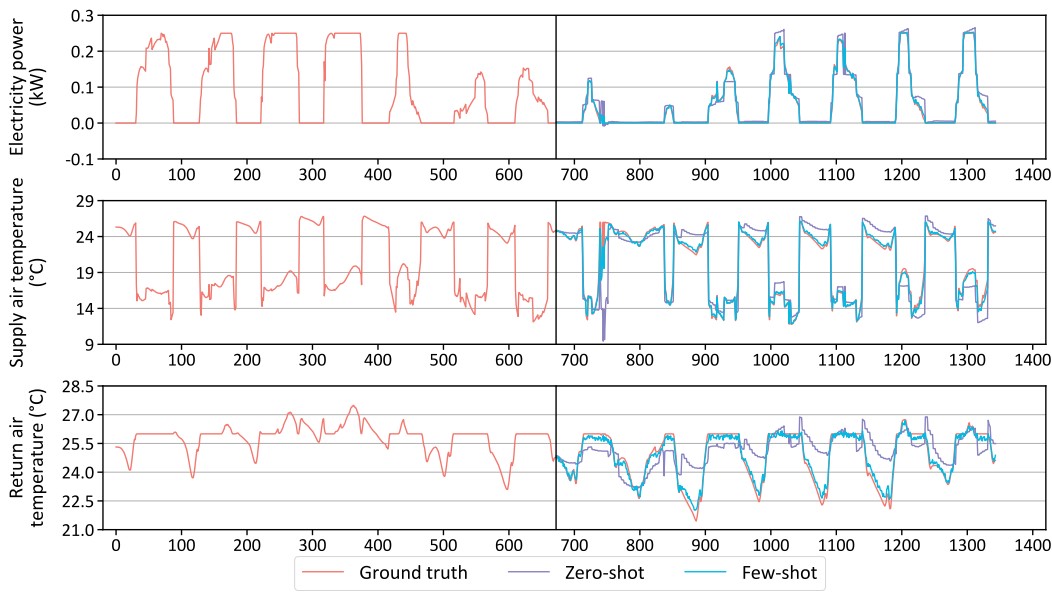

Figure 12: Visualization of time-series forecasting for key variables of a fan coil unit object, predicting the next 7 days based on the past 7 days. Predictions include zero-shot and few-shot (with one month of fine-tuning data).

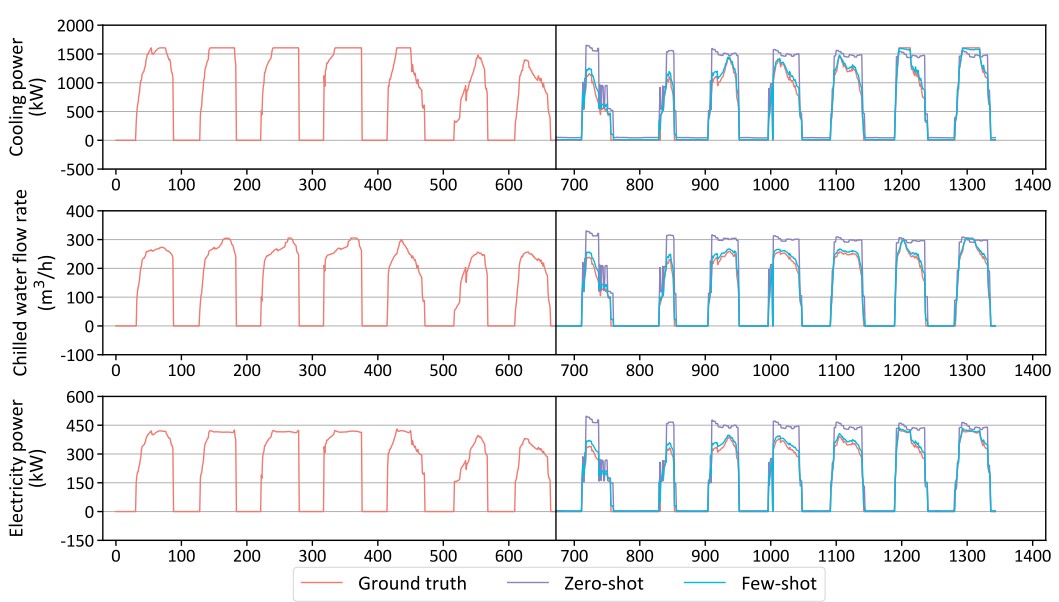

Figure 13: Visualization of time-series forecasting for key variables of a chiller system object, predicting the next 7 days based on the past 7 days. Predictions include zero-shot and few-shot (with one month of fine-tuning data).

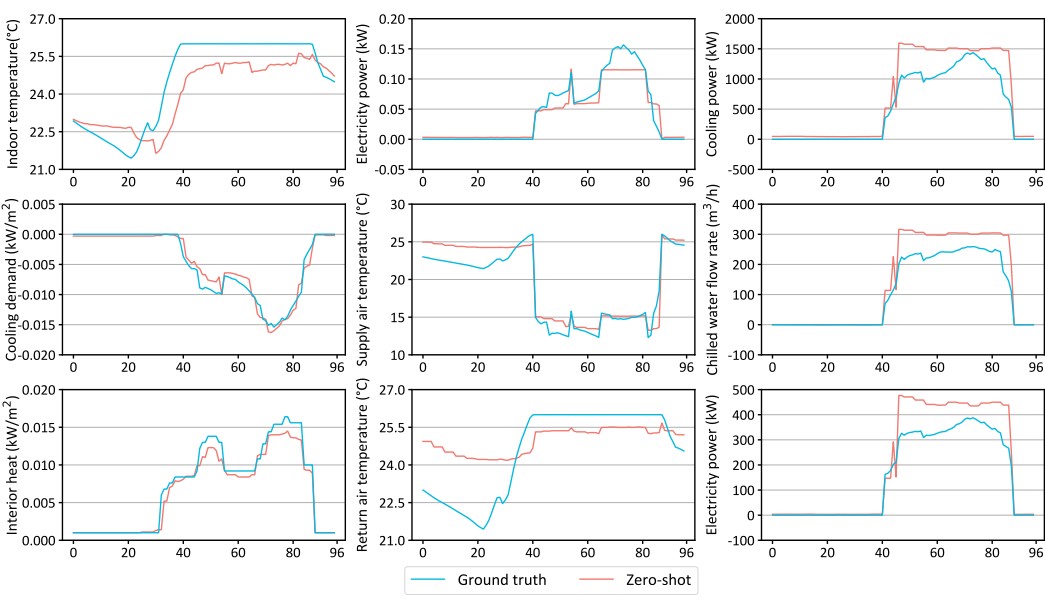

Figure 14: Curves for one day of three related object instances: Room, Fan Coil Unit, and Chiller System, along with their zero-shot prediction results. The left column shows three key variables for the Room object, the middle column for the Fan Coil Unit, and the right column for the Chiller System.

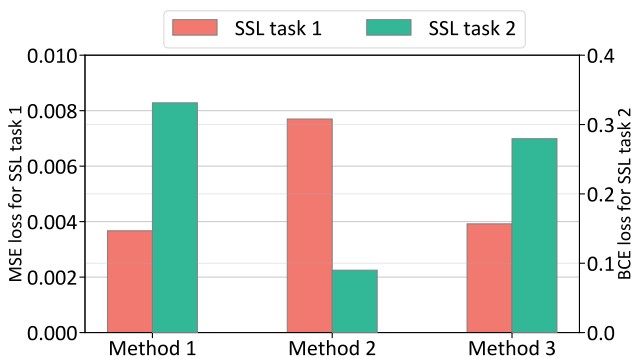

Figure 15: Comparison of three SSL training strategies: separate training, simultaneous training, and alternating training. Task 1 (masked time-series modeling) uses MSE loss, while Task 2 (masked edge modeling) uses BCE loss.

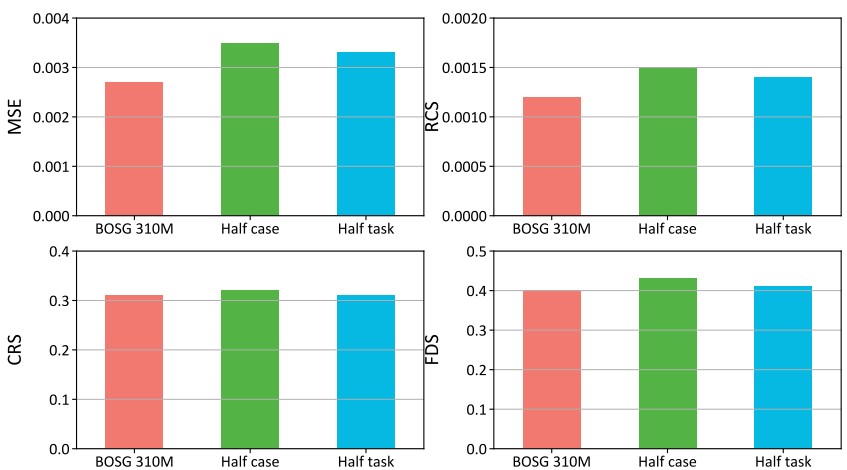

Figure 16: Performance comparison of the BOSG-310M model under three settings: full training, half the number of training cases, and half the number of tasks. Evaluation is based on multiple metrics.

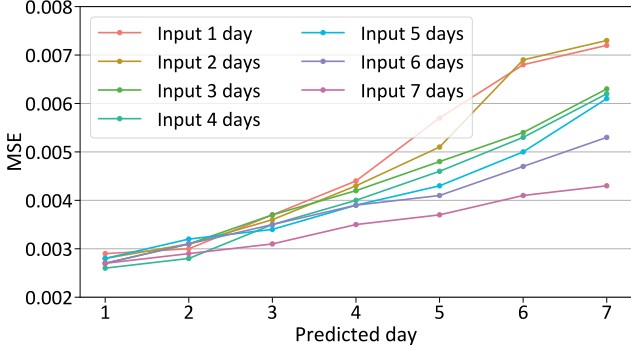

Figure 17: Prediction accuracy for daily rolling forecasts over a 7-day horizon. Each line represents a different input patch length, ranging from 1 to 7 days.

Table 15: Comparison of average relative errors among various normalization methods. The average error is calculated using the difference between the maximum and minimum values of the actual data for each instance variable as the base for CV-RMSE computation, excluding outlier instances such as devices that have been continuously inactive. The average CV-RMSE is computed for all instances of the same variable type.

| Object Type | Typical Variable | Normalization Method | | |
|---|---|---|---|---|
| | | Min-Max | Z-score | Multi-Instance |
| Room | Indoor Temperature | 2.1% | **2.0**% | 2.9% |
| | $CO_2$ | **2.4**% | 2.5% | 2.6% |
| Chiller | Chilled Water Supply Temperature | 7.9% | 8.1% | **6.8**% |
| | Chilled Water Return Temperature | 16.3% | 14.6% | **11.2**% |
| | Chilled Water Flow Rate | 45.6% | 33.9% | **16.8**% |
| Chilled Water Pump | Operating Power | 34.6% | 33.7% | **13.6**% |
| | Flow Rate | 39.9% | 38.2% | **12.8**% |
| Cooling Water Pump | Operating Power | 42.5% | 42.6% | **15.7**% |
| | Flow Rate | 45.5% | 48.4% | **16.1**% |
| Cooling Tower | Leaving Tower Water Temperature | 23.0% | 24.3% | **7.6**% |
| | Water Flow Rate | 35.3% | 33.8% | **15.5**% |
| Fan Coil Unit | Supply Air Temperature | 9.8% | 9.2% | **8.3**% |
| | Return Air Temperature | 5.7% | 6.1% | **3.6**% |
| Supply Air Fan | Fresh Air Flow Rate | 41.1% | 40.7% | **21.3**% |

in the fine-tuning phase, may deteriorate predictions for longer time horizons or when faced with highly divergent operational scenarios. To investigate this, we also tested fine-tuning with longer data windows (three months and six months), and found that, overall, predictions for extended timeframes benefited from the use of larger fine-tuning datasets. This suggests that a more extended fine-tuning period helps to mitigate overfitting and ensures better generalization for long-term predictions. However, a key challenge remains: how to fine-tune effectively with limited data while avoiding overfitting and preserving our foundation model's ability to learn the underlying physical dynamics. This continues to be an area of significant research interest.

We present daily profile curves for multiple dynamic variables of three different types of objects, along with their zero-shot prediction results. In Figure 14, we observe that the temporal relationships between the associated objects are effectively captured. In particular, the predictions for the chiller system and FCU demonstrate that the forecasted surge in cooling power for the Chiller System at time point 42 closely aligns with the predicted supply air temperature of the FCU at the same time. Similarly, the slight decrease in the room's indoor temperature at time point 66 is well-aligned with the small increase in the electricity power consumption of the FCU. In some instances, these temporal relationships are even more pronounced in the predicted data than in the actual observations, highlighting the model's capability to effectively capture interdependencies across various components in the system.

## J    ADDITIONAL RESULTS

### J.1    SELF-SUPERVISED LEARNING COMPARISON.

We investigate the interaction between two self-supervised learning (SSL) tasks: masked time-series modeling and masked edge prediction. Figure 15 compares three SSL training strategies: (1) training each task independently, (2) simultaneous multi-task training, and (3) our proposed alternating task training. Results show that while simultaneous training impairs the performance of the time-series task, the alternating training method maintains low loss for both tasks, offering a better trade-off between sequence forecasting and structural relation modeling.

Table 16: Time-series forecasting results on the Hydronic Domain dataset under three settings: pretrained zero-shot, pretrained few-shot, and no pre-training. Best results are in **bold**, second-best are underlined.

| Settings | Metric | LSTM | Autoformer | TFT | HTGNN | STD-MAE | TimesFM | MOIRAI | LLMTime | Time-LLM | HGTFT (Ours) |
|---|---|---|---|---|---|---|---|---|---|---|---|
| Zero-shot | MSE | 0.0066 | 0.0063 | 0.0046 | 0.0039 | 0.0038 | 0.0078 | 0.0073 | 0.0109 | 0.0093 | **0.0024** |
| | RCS | 0.0458 | 0.0260 | 0.0148 | 0.0084 | 0.0096 | 0.0315 | 0.0241 | 0.0346 | 0.0381 | **0.0013** |
| | CRS | 0.4812 | 0.5126 | 0.5457 | 0.3328 | **0.1949** | 0.3789 | 0.4730 | 0.6334 | 0.5462 | 0.2239 |
| | FDS | 0.4089 | 0.4989 | 0.5130 | 0.4619 | 0.4296 | 0.3334 | 0.3428 | 0.5223 | 0.4384 | **0.2922** |
| Few-shot | MSE | 0.0041 | 0.0035 | 0.0039 | 0.0032 | 0.0030 | 0.0036 | 0.0042 | 0.0062 | 0.0065 | **0.0017** |
| | RCS | 0.0344 | 0.0285 | 0.0136 | 0.0157 | 0.0120 | 0.0252 | 0.0195 | 0.0282 | 0.0285 | **0.0018** |
| | CRS | 0.3453 | 0.3587 | 0.3428 | 0.3079 | 0.2340 | 0.4287 | 0.3710 | 0.4202 | 0.3988 | **0.2280** |
| | FDS | 0.2643 | 0.2736 | 0.2916 | 0.2966 | 0.2960 | 0.2176 | 0.2499 | 0.3465 | 0.3052 | **0.1967** |
| No Pretrain | MSE | 0.0035 | 0.0039 | 0.0031 | 0.0036 | 0.0035 | 0.0043 | 0.0039 | 0.0062 | 0.0050 | **0.0019** |
| | RCS | 0.0326 | 0.0275 | 0.0130 | 0.0157 | 0.0190 | 0.0262 | 0.0197 | 0.0257 | 0.0264 | **0.0090** |
| | CRS | 0.3429 | 0.3486 | 0.3489 | 0.3069 | 0.2361 | 0.3648 | 0.3993 | 0.3957 | 0.3641 | **0.2322** |
| | FDS | 0.2928 | 0.3307 | 0.2556 | 0.3377 | 0.3092 | 0.2493 | **0.2310** | 0.4098 | 0.3020 | 0.2716 |

Table 17: Time-series forecasting results on the Airflow / HVAC Domain Domain dataset under three settings: pretrained zero-shot, pretrained few-shot, and no pre-training. Best results are in **bold**, second-best are underlined.

| Settings | Metric | LSTM | Autoformer | TFT | HTGNN | STD-MAE | TimesFM | MOIRAI | LLMTime | Time-LLM | HGTFT (Ours) |
|---|---|---|---|---|---|---|---|---|---|---|---|
| Zero-shot | MSE | 0.0054 | 0.0051 | 0.0038 | 0.0033 | 0.0035 | 0.0069 | 0.0060 | 0.0092 | 0.0079 | **0.0022** |
| | RCS | 0.0386 | 0.0227 | 0.0120 | 0.0074 | 0.0084 | 0.0278 | 0.0216 | 0.0306 | 0.0327 | **0.0010** |
| | CRS | 0.4028 | 0.4567 | 0.4422 | 0.2953 | 0.1870 | 0.3401 | 0.4071 | 0.5263 | 0.4750 | **0.1825** |
| | FDS | 0.3542 | 0.4452 | 0.4207 | 0.4023 | 0.3598 | 0.3084 | 0.2545 | 0.4547 | 0.3935 | **0.2499** |
| Few-shot | MSE | 0.0036 | 0.0026 | 0.0025 | 0.0028 | 0.0024 | 0.0029 | 0.0035 | 0.0051 | 0.0058 | **0.0015** |
| | RCS | 0.0283 | 0.0242 | 0.0112 | 0.0132 | 0.0098 | 0.0208 | 0.0168 | 0.0250 | 0.0235 | **0.0016** |
| | CRS | 0.2845 | 0.2987 | 0.2792 | 0.2579 | 0.2035 | 0.3656 | 0.3076 | 0.3395 | 0.3366 | **0.1992** |
| | FDS | 0.2170 | 0.2288 | 0.2376 | 0.2632 | 0.2480 | 0.1941 | 0.2055 | 0.3048 | 0.2593 | **0.1599** |
| No Pretrain | MSE | 0.0030 | 0.0033 | 0.0025 | 0.0029 | 0.0029 | 0.0036 | 0.0034 | 0.0051 | 0.0044 | **0.0018** |
| | RCS | 0.0263 | 0.0235 | 0.0116 | 0.0136 | 0.0162 | 0.0231 | 0.0158 | 0.0231 | 0.0213 | **0.0081** |
| | CRS | 0.2900 | 0.2797 | 0.3093 | 0.2711 | 0.2180 | 0.3053 | 0.3300 | 0.3403 | 0.3143 | **0.2029** |
| | FDS | 0.2479 | 0.2646 | 0.2288 | 0.2772 | 0.2534 | 0.2335 | 0.2446 | 0.3467 | 0.2692 | **0.2278** |

## J.2 Impact of Task Diversity and Data Quantity.

Leveraging SSL-pretrained weights, we adopt a sequential multi-task learning framework where downstream tasks are optimized one after another. During training, the average task loss consistently decreases across rounds, and the rate of change stabilizes, indicating convergence under the serialized learning schedule. We further conduct ablation studies by halving the number of training tasks and the number of training cases, respectively. As shown in Figure 16, both reductions lead to moderate performance degradation, highlighting the importance of maintaining sufficient task diversity and data coverage for robust generalization.

## J.3 Effect of Input Patch Length and Forecasting Horizon.

We evaluate the model's performance across different input and output durations, ranging from 1 to 7 days. Figure 17 presents the results of daily rolling forecasts, where the x-axis denotes the target prediction day and the y-axis represents the corresponding MSE. Each curve corresponds to a different input patch length. The results demonstrate that longer input sequences generally yield improved accuracy, particularly for longer forecasting horizons.

## J.4 Multiphysics Decomposition of Building Systems and Cross-Domain Forecasting Results

Although our experiments are conducted in the building domain, this environment is inherently multiphysics, consisting of several interacting sub-domains. A modern building comprises numerous subsystems, each governed by distinct physical principles and involving heterogeneous object

Table 18: Time-series forecasting results on the Thermal Envelope Domain dataset under three settings: pretrained zero-shot, pretrained few-shot, and no pre-training. Best results are in **bold**, second-best are underlined.

| Settings | Metric | LSTM | Autoformer | TFT | HTGNN | STD-MAE | TimesFM | MOIRAI | LLMTime | Time-LLM | HGTFT (Ours) |
|---|---|---|---|---|---|---|---|---|---|---|---|
| Zero-shot | MSE | 0.0125 | 0.0115 | 0.0085 | 0.0076 | 0.0078 | 0.0133 | 0.0130 | 0.0197 | 0.0186 | **0.0047** |
| | RCS | 0.0766 | 0.0495 | 0.0260 | 0.0147 | 0.0165 | 0.0616 | 0.0441 | 0.0603 | 0.0641 | **0.0022** |
| | CRS | 0.8594 | 0.8815 | 0.9421 | 0.6541 | **0.3721** | 0.7361 | 0.9285 | 1.0628 | 1.0600 | 0.3925 |
| | FDS | 0.7280 | 0.8681 | 0.8573 | 0.8716 | 0.7755 | 0.7171 | 0.5979 | 1.0270 | 0.7797 | **0.4871** |
| Few-shot | MSE | 0.0071 | 0.0053 | 0.0055 | 0.0056 | 0.0057 | 0.0070 | 0.0072 | 0.0104 | 0.0112 | **0.0029** |
| | RCS | 0.0656 | 0.0520 | 0.0251 | 0.0264 | 0.0237 | 0.0465 | 0.0367 | 0.0535 | 0.0482 | **0.0030** |
| | CRS | 0.6572 | 0.7032 | 0.6572 | 0.5697 | **0.3595** | 0.7748 | 0.7198 | 0.7372 | 0.7755 | 0.4398 |
| | FDS | 0.5228 | 0.5040 | 0.4999 | 0.5026 | 0.5515 | 0.4346 | 0.4334 | 0.6498 | 0.5284 | **0.3595** |
| No Pretrain | MSE | 0.0069 | 0.0066 | 0.0054 | 0.0055 | 0.0059 | 0.0079 | 0.0074 | 0.0112 | 0.0091 | **0.0033** |
| | RCS | 0.0600 | 0.0476 | 0.0246 | 0.0282 | 0.0320 | 0.0501 | 0.0369 | 0.0491 | 0.0491 | **0.0181** |
| | CRS | 0.6514 | 0.6875 | 0.5858 | 0.5388 | 0.4794 | 0.6665 | 0.6798 | 0.7483 | 0.6874 | **0.4525** |
| | FDS | 0.5668 | 0.5731 | 0.5054 | 0.6219 | 0.5356 | 0.4585 | **0.4299** | 0.7306 | 0.5622 | 0.5195 |

Table 19: Time-series forecasting results on the Refrigeration / Plant Domain dataset under three settings: pretrained zero-shot, pretrained few-shot, and no pre-training. Best results are in **bold**, second-best are underlined.

| Settings | Metric | LSTM | Autoformer | TFT | HTGNN | STD-MAE | TimesFM | MOIRAI | LLMTime | Time-LLM | HGTFT (Ours) |
|---|---|---|---|---|---|---|---|---|---|---|---|
| Zero-shot | MSE | 0.0102 | 0.0103 | 0.0074 | 0.0067 | 0.0075 | 0.0129 | 0.0126 | 0.0166 | 0.0163 | **0.0041** |
| | RCS | 0.0638 | 0.0448 | 0.0240 | 0.0133 | 0.0142 | 0.0551 | 0.0392 | 0.0537 | 0.0627 | **0.0020** |
| | CRS | 0.7575 | 0.7447 | 0.8515 | 0.5506 | 0.3466 | 0.6324 | 0.7807 | 0.9340 | 0.9009 | **0.3393** |
| | FDS | 0.6069 | 0.7913 | 0.7669 | 0.7728 | 0.6959 | 0.6359 | 0.5280 | 0.9461 | 0.7022 | **0.4505** |
| Few-shot | MSE | 0.0058 | 0.0047 | 0.0044 | 0.0047 | 0.0054 | 0.0062 | 0.0068 | 0.0084 | 0.0104 | **0.0027** |
| | RCS | 0.0560 | 0.0470 | 0.0212 | 0.0224 | 0.0199 | 0.0407 | 0.0337 | 0.0437 | 0.0407 | **0.0026** |
| | CRS | 0.6011 | 0.6550 | 0.6379 | 0.4607 | **0.3331** | 0.7084 | 0.6494 | 0.6822 | 0.6688 | 0.3730 |
| | FDS | 0.4422 | 0.4004 | 0.4726 | 0.4111 | 0.4893 | 0.3848 | 0.3806 | 0.5345 | 0.4996 | **0.3027** |
| No Pretrain | MSE | 0.0065 | 0.0057 | 0.0045 | 0.0050 | 0.0056 | 0.0069 | 0.0061 | 0.0094 | 0.0082 | **0.0031** |
| | RCS | 0.0556 | 0.0402 | 0.0219 | 0.0239 | 0.0273 | 0.0417 | 0.0313 | 0.0465 | 0.0433 | **0.0148** |
| | CRS | 0.5184 | 0.5905 | 0.5669 | 0.4500 | **0.3206** | 0.6289 | 0.6133 | 0.6671 | 0.6154 | 0.4650 |
| | FDS | 0.4610 | 0.4556 | 0.4504 | 0.5027 | 0.4599 | 0.4176 | 0.4249 | 0.6336 | 0.4770 | **0.4002** |

types. For clarity, we decompose the building system into four canonical sub-domains, as commonly recognized in building science and HVAC engineering.

1. **Hydronic Domain Representative objects:** pumps, valves, tanks, hydronic loops, and water distribution networks
   **Governing physics:** fluid dynamics, mass conservation, hydraulic balance

2. **Airflow / HVAC Domain Representative objects:** fans, ducts, dampers, variable air volume (VAV) components, heat exchangers
   **Governing physics:** airflow mechanics, convective heat transfer, pressure–flow coupling

3. **Thermal Envelope Domain Representative objects:** walls, windows, shading devices, indoor zones, outdoor environment
   **Governing physics:** conduction, radiation, heat storage, thermodynamic balance

4. **Refrigeration / Plant Domain Representative objects:** chillers, compressors, cooling towers, condensers, evaporators
   **Governing physics:** vapor compression cycles, phase-change thermodynamics, energy balance

These sub-domains collectively span multiple physical fields—including heat transfer, fluid flow, thermodynamics, mechanical work, and cyber–physical control—and exhibit cross-physics and cross-entity couplings. Demonstrating consistent forecasting performance across these heterogeneous components provides evidence that our framework is not restricted to a single physical mechanism but instead supports general **multi-entity, multi-variable forecasting with heterogeneous interactions**.

To further illustrate the generality of our approach, Tables 16 to 19 present the forecasting results across the four canonical sub-domains. The tables report evaluation metrics under three settings: **no pre-training**, **zero-shot performance**, and **few-shot fine-tuning**. This experiment follows a setup similar to that of Table 2 in the main text; however, it is conducted on the MBS dataset—using 50 randomly sampled buildings for testing—because the BTS dataset does not provide full or consistent coverage of all four sub-domains.

## J.5  PERFORMANCE GAINS UNDER FIXED CAPACITY: JUSTIFYING MODEL COMPLEXITY

To assess when a complex architecture like HGTFT is justified compared to simpler models such as LSTM, we conduct two sets of comparative experiments:

1. **Scenario 1: Single-variable prediction** We use a univariate time-series forecasting dataset (ETT) to evaluate performance when only a single object type is involved.

2. **Scenario 2: Multi-object prediction** We construct a multi-object scenario containing two object types, each with three information channels (two dynamic variables and one static feature), to evaluate the benefits of HGTFT in capturing cross-entity interactions.

Table 20 and Table 21 present the corresponding forecasting results for the two scenarios. This experimental setup allows us to characterize practical trade-offs: while HGTFT and LSTM perform comparably in the single-object scenario with limited interactions, HGTFT demonstrates clear advantages in the multi-object setting, benefiting from its graph–temporal fusion and ability to model heterogeneous interactions. These results highlight the regimes where a more complex architecture is warranted, and when simpler models suffice.

Table 20: Comparison of HGTFT and LSTM under different model sizes (Scenario 1). FLOPS are reported in GFLOPS ($10^9$ FLOPS).

| Parameters | Embedding Dim | HGTFT GFLOPS | HGTFT MSE | LSTM GLOPS | LSTM MSE |
|---|---|---|---|---|---|
| 0.8M | 64 | 2,108 | 0.683 | 3,494 | 0.694 |
| 3M | 128 | 16,369 | 0.521 | 13,753 | 0.607 |
| 10M | 256 | 128,965 | 0.413 | 54,561 | 0.593 |

Table 21: Comparison of HGTFT and LSTM under different model sizes (Scenario 2). FLOPS are reported in GFLOPS ($10^9$ FLOPS).

| Parameters | Embedding Dim | HGTFT GFLOPS | HGTFT MSE | LSTM GFLOPS | LSTM MSE |
|---|---|---|---|---|---|
| 1.5M | 64 | 4,217 | 0.0097 | 6,989 | 0.0162 |
| 6M | 128 | 32,738 | 0.0056 | 27,505 | 0.0149 |
| 20M | 256 | 257,930 | 0.0025 | 109,121 | 0.0144 |

## K  LIMITATIONS AND FUTURE WORK

Despite the promising results, this work still faces several limitations and open challenges:

1. **Generality across physical domains:** Our experiments focus on building operation systems as representative multiphysics environments, capturing rich interactions among thermal, hydraulic, and control processes. Future work can extend validation to other complex physical systems (e.g., energy grids or manufacturing processes) to further establish and demonstrate the generality of the proposed framework.

2. **Dataset coverage:** Public datasets for multiphysics forecasting remain limited. While our MBS dataset is larger and more comprehensive than prior resources, expanding it to include additional object types, physical processes, and control scenarios would further enhance its representativeness and support broader evaluation, fostering community progress.

3. **Few-shot adaptation:** While few-shot finetuning yields clear benefits under short horizons or near-distribution conditions, its performance can degrade when forecasting over longer horizons or under substantial distributional shifts. Developing selective adaptation strategies that automatically identify which parameters or modules to adapt will be crucial for improving robustness in such settings.

Addressing these challenges will strengthen the robustness, flexibility, and scalability of multi-physics forecasting, paving the way for broader deployment in real-world complex physical systems.

## L    LLM USAGE

Large language models (LLMs) were used solely for grammar correction and stylistic refinement of the manuscript. They did not contribute to research ideation, model design, data analysis, or experimental results. The authors take full responsibility for the scientific content of the paper.

