# OpenReview forum: "Heterogeneous Graph Temporal Fusion Transformer for Time Series Forecasting in Multi-Domain Physical Systems"
_ICLR.cc/2026/Conference — Submitted to ICLR 2026_

### Official Review · Reviewer_W4Fj · 2025-10-31

**Soundness:** 3
**Presentation:** 2
**Contribution:** 3
**Rating:** 4
**Confidence:** 3

**Summary:**

This paper proposes the Heterogeneous Graph Temporal Fusion Transformer (HGTFT), a pre-training and fine-tuning framework designed for time series forecasting in multi-domain physical systems. The HGTFT model tokenizes observation points and uses a graph-temporal architecture to integrate heterogeneous static and dynamic information. The framework also introduces a novel Multi-Instance Normalization technique and a multi-stage training pipeline. The main findings suggest that HGTFT achieves strong zero-shot and few-shot performance, outperforming various baselines in both prediction accuracy and physical plausibility.

**Strengths:**

1. This paper addresses forecasting in complex real-world physical systems, which is a challenging and valuable problem. Furthermore, the focus on multi-domain interactions and physical consistency represents an important research area.
2. The authors conducted extensive ablation studies on the model architecture, provided a detailed analysis of model scalability, and compared different training strategies and normalization methods.
3. The introduction and release of the Multiphysics Building System (MBS) dataset is a valuable contribution to the community, providing a large-scale, complex benchmark dataset for this research field.

**Weaknesses:**

1. The baselines compared in the experiment are weak. The authors should compare against more recent and SOTA models, such as the deep learning models PatchTST [1], DLinear [2], and TQNet [3], as well as time series foundation models like Sundial [4].
2. The RCS introduced in the paper is not a general or objective evaluation metric. It is essentially a set of heuristics defined by the authors themselves. The authors use these rules as a loss function during the training stage and then use them again during the evaluation stage to demonstrate the model's physical consistency. Therefore, the model will naturally perform better on the RCS metric compared to the baselines. However, this does not fully prove that the model truly understands the physics; it only proves that it has learned to adhere to these hard-coded rules.
3. The scalability of the proposed architecture appears to be poor. According to the Scaling Study in Table 7, increasing the model parameter count from 310M to 1.26B results in a minimal improvement in the primary MSE metric (from 0.0027 to 0.0025). Meanwhile, the FDS (Frequency Domain Similarity) metric actually worsens (increasing from 0.405 to 0.416).

[1] A Time Series is Worth 64 Words: Long-term Forecasting with Transformers. (ICLR 2023)

[2] Are Transformers Effective for Time Series Forecasting? (AAAI 2023)

[3] Temporal Query Network for Efficient Multivariate Time Series Forecasting. (ICML 2025)

[4] Sundial: A Family of Highly Capable Time Series Foundation Models. (ICML 2025)

**Questions:**

1. For the channel-independent baselines in the experiment, how was the pre-training or few-shot learning conducted? Were the models trained using only the target variable, or were all variables used in the training process?
2. Is the framework truly generalizable? The paper only conducts sufficient experiments in a single physical domain (the building domain). Would migrating it to other domains require significant effort (e.g., redesigning the RCS loss), and what kind of performance changes could be expected?
3. In Appendix A, the authors use a simplified FCU physical model as an example. They first derive differential equations, use them to generate simulated data, and then show HGTFT can perfectly fit this data. However, the inputs in this example are perfect sine waves. Successfully fitting this toy problem does not prove that HGTFT understands the physics. Is there any demonstration on truly complex and chaotic real-world data?
4. According to Appendix B, the simulated data contains 600B time points, while the real project data has only 16B time points. Does this imply that the pre-trained model spends the vast majority of its time learning the simulator's behavior? It is possible the model is merely overfitting to the simulator's simplified rules and specific dynamics rather than learning the truly complex physical laws of the real world.

---

> ### Author Response · Authors · 2025-11-20
>
> We appreciate the reviewer’s insightful feedback and provide the following responses to address the concerns raised.
>
> **Response to Weakness 1:**
>
> We appreciate the suggestion to include additional recent time-series forecasting baselines. However, the methods cited (PatchTST, DLinear, TQNet, Sundial) are designed for pure sequence modeling and do not account for heterogeneous entities, multi-object interactions, or graph topology. Our problem setting is heterogeneous-object graph time-series forecasting, where incorporating structural relationships and static attributes is essential.
>
> In this context, we include LSTM and Transformer-based models (e.g., Autoformer) primarily to provide a clear contrast between pure temporal models—which cannot utilize graph topology—and models capable of leveraging heterogeneous graph structure such as HGTFT. More extensive comparisons with spatial–temporal forecasting models are already provided in Table 13 of the Appendix G, where we evaluate against a broad set of ST baselines that are more appropriate for our task.
>
>
> **Response to Weakness 2:**
>
> We agree: RCS is a heuristic metric, used both as a training loss and evaluation criterion, so HGTFT’s superior RCS performance over baselines is expected.
>
> The primary function of RCS:
>
> **RCS' function is to provide domain-informed regularization.** The constraints encoded in RCS follow established engineering rules derived from real building and HVAC systems, as detailed in Appendix F.4. Instead of enforcing exact governing equations, **RCS smooths the optimization landscape and steers the model away from physically implausible regions**. This helps stabilize training and improves predictive accuracy. For example, Table 2 shows that incorporating RCS reduces MSE from 0.0040 to 0.0033, and in our few-shot experiments it consistently brings 10–18% improvement by reducing the effective hypothesis space. The training objective maintains MSE as the principal accuracy term while gradually increasing the weight of RCS to enhance consistency. The complete staged schedule is summarized in Table 11, with implementation details provided in Section F.7.
>
> Clarification on Physical Understanding:
>
> We do not claim true physics understanding. As stated throughout, HGTFT enhances temporal understanding, relational understanding, and physical plausibility through graph-temporal fusion and physics-aware training. RCS should be viewed as a practical, domain-aware regularizer—analogous to classical smoothness priors, but grounded in engineering knowledge. Its value lies in enabling better generalization and more stable predictions in heterogeneous multi-entity systems where full PDE-based modeling is infeasible.
>
>
> **Response to Weakness 3:**
>
> We appreciate the reviewer’s observation. The scaling results in Table 7 (Table 8 in the revised version) reflect an inherent predictability limit in complex physical systems. On the MBS dataset—containing coupled thermal, hydronic, and electrical dynamics—the 310M HGTFT already captures nearly all learnable structure, leaving limited signal for a 1.26B model to exploit.
>
> Additional capacity cannot overcome irreducible uncertainty arising from unmeasured disturbances, sensor noise, and regime-dependent interactions, which bound achievable MSE. The slight FDS degradation may occur because larger models can begin to fit high-frequency variations driven by noise rather than true dynamics, to which FDS is particularly sensitive.
>
> Overall, the results indicate saturation in the data—not a scalability issue of the architecture—and 300M-scale models are sufficient for this domain.
>
> **Response to Question 1:**
>
> For the channel-independent baselines, we adopt the same pre-training and few-shot learning pipeline as described in Section 5, except that these models do not include graph fusion or any cross-entity aggregation.
>
> Regarding the variables used for training, all baselines were trained per object type, using all dynamic variables associated with that object type. For each forecasting task, the input–output specification of the dynamic variables is kept fully consistent with the HGTFT setup.
>
> TFT:
> TFT leverages both the object type’s full set of dynamic variables and its static features during pre-training and few-shot fine-tuning.
>
> Other channel-independent baselines (e.g., LSTM, Autoformer):
> These methods use only the dynamic variables of each object type. They cannot incorporate graph information or static features due to architectural limitations.

---

> > ### Author Response · Authors · 2025-11-20
> >
> > **Response to Question 2:**
> >
> > Yes, the framework is generalizable. While our experiments focus on the building domain, **we explicitly decompose it into four canonical sub-domains** —hydronic systems, HVAC/airflow, thermal envelope, and refrigeration/plant systems. Each sub-domain is governed by distinct physical principles, and together they span fluid dynamics, heat transfer, thermodynamics, mechanical work, and control dynamics.
> >
> > We validate HGTFT’s performance separately on each sub-domain (**Appendix J.4** in revised paper), demonstrating consistent accuracy and relational/temporal modeling, which provides strong evidence that the framework can generalize across heterogeneous physical mechanisms.
> >
> > Cross-domain migration: Most components—tokenization, heterogeneous graph construction, temporal modeling, and the pretraining–finetuning pipeline—transfer directly to new physical domains without modification. Only the RCS loss requires redesign to encode domain-specific soft constraints. This process is analogous to adapting priors or regularizers when moving to a new physics domain and does not require changes to the model architecture.
> >
> > Expected performance: Without modifying RCS, the model still generalizes effectively due to its graph–temporal representations. With domain-specific RCS, further improvements in predictive accuracy and physical plausibility are expected, similar to the observed gains in building sub-domains.
> >
> > In summary, the explicit sub-domain decomposition and separate evaluation provide concrete evidence of generalization, and the framework is readily transferable to other physical domains with minimal adaptation.
> >
> >
> > **Response to Question 3:**
> >
> > Our validation strategy progresses from simple examples to prove convergence analytically, to complex real-world cases where closed-form solutions are infeasible.
> >
> > - Appendix A.1 provides a simple FCU example to validate convergence on a known dynamical system.
> >
> > - Appendix A.2 extends this to a more complex, nonlinear multi-zone HVAC model with coupled differential-algebraic equations, stochastic disturbances, and irregular forcing.
> >
> > Most validation, is conducted on real-world, chaotic building data:
> >
> > - BTS dataset (NeurIPS 2024; Prabowo et al., 2024): Real collected from three operational buildings with sensor noise, drift, missing samples, faults, and non-stationary interventions.
> >
> > - MBS dataset (Appendix B): Includes 16B real project time points from 1,045 real-world buildings, capturing unmodeled dynamics (e.g., occupant behavior, weather extremes, equipment degradation).
> >
> > These sources introduce true chaos and complexity: partial observability, time-varying couplings, and emergent failures. On BTS (unseen during pre-training), HGTFT achieves zero-shot and few-shot results in Table 2, with extensive ablation validation on MBS in Table 4, demonstrating robust generalization to real, chaotic multi-physics environments.
> >
> >
> > **Response to Question 4:**
> >
> > We agree that the real-world dataset is much smaller than the simulated data. Our use of simulation is not intended to replace real physics, but to augment the coverage of training scenarios where real data is limited.
> >
> > - Real data as foundation: The 1,045-building dataset we collected remains one of the largest real-world HVAC datasets, but insufficient alone for pre-training a large model. Hence, simulation is used to expand temporal coverage and diversity.
> >
> > - Physics-based simulation: The synthetic data is generated from EnergyPlus, a widely validated physics engine based on thermodynamics, heat transfer, and fluid flow. This ensures the model primarily learns physically plausible operational patterns rather than arbitrary or nonphysical behaviors.
> >
> > - Cross-dataset generalization: To prevent overfitting to simulator-specific dynamics, simulation templates are derived from 12 diverse real buildings and calibrated to match realistic system behavior. Real data is injected more frequently in later training stages, ensuring updates remain anchored to real physical behavior. Importantly, the pretrained model achieves strong performance on the external BTS dataset—whose buildings differ substantially from MBS in climate, occupancy, and HVAC configurations—demonstrating cross-domain generalization rather than mere memorization of simulator rules.
> >
> > - Clarification on physics learning: The model does not explicitly recover first-principles physics. Instead, it captures high-dimensional statistical patterns that reflect the operational manifestations of physical laws encoded in the data. These patterns enable accurate forecasting, approximate conservation behavior, and causal consistency, but should not be interpreted as direct learning of governing equations.
> >
> >
> > We hope our clarifications address the reviewer’s concerns.

---

### Official Review · Reviewer_Bu3n · 2025-10-31

**Soundness:** 3
**Presentation:** 3
**Contribution:** 2
**Rating:** 6
**Confidence:** 2

**Summary:**

This paper proposes HGTFT, a transformer framework for forecasting in heterogeneous multi-domain physical systems. It fuses static and dynamic variables in a heterogeneous graph, integrates temporal attention with relation-specific aggregation. Evaluations show substantial improvements over baselines and strong zero/few-shot transfer across realistic multiphysics systems.

**Strengths:**

1. the topic is timely and valuable.

2. The paper clearly defines the new setting of heterogeneous graph forecasting in multi-domain physical systems, extending beyond conventional data.

3. The proposed graph-temporal fusion with physics-aligned losses is technically well motivated and addresses both accuracy and physical consistency, which many data-driven models ignore

4. Comprehensive experiments across synthetic and real-world datasets support the method’s claims

**Weaknesses:**

1. The technical increment over existing graph transformers or physics-informed forecasting models is limited. This paper introduced: a heterogeneous graph encoder, a temporal transformer, and physics-inspired regularizers. However, each individual piece has been seen in earlier spatiotemporal or physics-informed learning work. The paper’s novelty lies more in integration and application to building energy systems than in a new architectural mechanism.

2. The proposed physics-informed losses (RCS/CRS/FDS) are heuristic rather than derived from governing equations. The extent to which they enforce true physical constraints?

**Questions:**

1. Table 2 shows large RCS improvements (e.g., 0.0158 → 0.0018 zero shot); could the authors how this metric generalizes to unseen physics domains?

2. In Section 5.3, the weighting of the four loss terms (MSE, RCS, CRS, FDS) appears fixed; could the authors show how performance changes when these weights are learned or tuned, to verify robustness?

3. The multi-instance normalization in Eq. (9) aggregates across percentile bounds. Could the authors explain how this compares against standard per-feature normalization across node types?

---

> ### Author Response · Authors · 2025-11-20
>
> We appreciate the reviewer’s insightful feedback and provide the following responses to address the concerns raised.
>
> **Response to Weakness 1:**
>
> While each individual component of our method has appeared in prior work, their isolated use is precisely why they have been widely adopted in spatiotemporal and physics-informed learning. However, to our knowledge, no prior study has effectively integrated the full range of information present in our problem setting—heterogeneous entities, heterogeneous relations, and mixed static–dynamic attributes—into a unified architecture that yields practical value in real-world multi-physics forecasting. More concretely, prior methods provide only partial coverage: TFT (Lim et al., 2021) introduces static–dynamic feature fusion but is restricted to single-object forecasting; HTGNN (Fan et al., 2022) models heterogeneous temporal graphs, yet does not incorporate heterogeneous static and dynamic attributes of entities, and focuses solely on node-type heterogeneity rather than the richer relation heterogeneity.
>
> Our contribution is not a simple juxtaposition of existing modules, but the result of extensive design exploration leading to an architecture where the components work in a coherent and complementary manner. This includes:
>
> - Token design, where each token represents the complete state of an entity—its static attributes and dynamic features—at a given time step.
>
> - Framework-level modeling choices, where we explicitly differentiate edge relations and allocate distinct parameter sets for each relation type. This captures finer-grained dependencies and avoids semantic entanglement between heterogeneous physical or logical interactions (Eq. 6–7).
>
> - Architectural trade-offs, such as the placement and depth of temporal and graph layers (Appendix D), and the observed benefit of inserting a temporal Transformer layer before the heterogeneous graph layer (Table 6).
>
> - Improvements in normalization, which are essential for achieving balanced multi-variable performance, as discussed in our response to Question 3.
>
> In summary, our work targets a class of problems that is both underrepresented in the time-series forecasting literature and pervasive in real-world settings: multi-object, multi-physics systems with heterogeneous entities, heterogeneous relations, and mixed static–dynamic information. The integrated design tailored for these scenarios constitutes the main technical contribution of this paper.
>
> **Response to Weakness 2 and Question 1:**
>
> We agree that RCS/CRS/FDS are heuristic soft constraints rather than physics derived from governing equations. They are not intended to serve as evidence of deep physical reasoning. Instead, **they function as domain-informed regularizers that bias the pretrained model away from implausible operating regimes in multi-entity systems**. Consistent 10–18% MSE improvements across pre-training and few-shot fine-tuning indicate that these losses primarily enhance temporal and relational consistency rather than enforce hard physical laws.
>
> Importantly, the HGTFT architecture is entirely independent of these heuristic losses. The model transfers across domains without any reliance on system-specific physics, while RCS/CRS/FDS remain optional add-ons that do not modify the architecture or training pipeline.
>
> **Regarding generalization**, the constraints encoded by RCS were intentionally designed to be system-agnostic. They reflect broad physical plausibility principles—such as state bounds, resource consistency, and cross-component coupling—that hold across diverse building subsystems rather than being tied to any particular topology or device configuration. This is why large RCS improvements in Table 2 occur on BTS, even though BTS differs substantially from the pre-training corpus (MBS) in climate, control strategies, and operating patterns. **Additional zero/few-shot results in Appendix J.4 (in revised paper) further demonstrate consistent gains across four distinct physical sub-domains (hydronic loops, HVAC air systems, thermal envelope behavior, and refrigeration), suggesting that these soft constraints generalize well across unseen physics combinations.**
>
> Conceptually, RCS acts as a soft physical plausibility prior: unlike PINNs, it does not require explicit PDE specification and thus does not rigidly couple the model to a known set of equations. This flexibility makes it more suitable for heterogeneous, partially observed multi-entity systems where a complete physics specification is infeasible. For new domains that introduce additional physical behaviors, new constraint terms can be added modularly without altering the pretrained architecture.

---

> > ### Author Response · Authors · 2025-11-20
> >
> > **Response to Question 2:**
> >
> > Supervised learning is performed on top of a foundation model pretrained with self-supervised objectives. The supervised loss-weight schedule is designed to gradually shift optimization from accuracy to physical consistency: early stages prioritize MSE to quickly adapt task-specific parameters, while later stages progressively increase the weights of RCS/CRS/FDS to enhance physical plausibility and generalization. This staged strategy (Table 11) ensures stable optimization; details are provided in Section F.7.
> >
> > We experimented with automatic loss-weight learning methods, such as DWA, but found training highly unstable. Dynamic adjustment can amplify short-term fluctuations in loss values, particularly when losses differ in scale or convergence speed, destabilizing gradient updates and hindering convergence.
> >
> > We also tried introducing additional losses beyond MSE from the first stage. However, with multiple objectives and a large parameter search space, optimization became difficult and often failed to converge.
> >
> > To address this, we adopt a staged weighting strategy: after pretraining with MSE to reach a reasonable accuracy range (less than 0.01), physical-consistency losses (e.g., RCS) are gradually introduced. MSE remains the dominant component throughout (no less than 50%), ensuring prediction accuracy is prioritized. RCS weight is progressively increased in later stages, up to roughly 50%. Minor adjustments (e.g., stage 5 set to 0.6 MSE and 0.4 RCS) produce negligible differences (<1% across metrics). Following this principle, the model consistently achieves accurate and physically plausible predictions.
> >
> > **Response to Question 3:**
> >
> > Due to space limitations, the detailed comparison of normalization methods is provided in Appendix H. In our multi-physics, multi-entity setting, predictive accuracy reflects the joint performance across many interacting objects and variables, rather than improvements for individual features alone. As shown in Table 15, our proposed Multi-Instance Normalization significantly improves overall performance, particularly for variables where other normalization methods yield poor predictions, demonstrating a more balanced and robust treatment across heterogeneous entities.
> >
> > We hope that the explanations and added experiments help to address the reviewer’s concerns.

---

### Official Review · Reviewer_uK4E · 2025-11-01

**Soundness:** 2
**Presentation:** 3
**Contribution:** 2
**Rating:** 4
**Confidence:** 3

**Summary:**

The paper introduces HGTFT, a graph transformer architecture for time series forecasting in multi-physics domains. This approach improves upon drawbacks of existing methods in multi-physics settings, which often struggle to perform under the complexity of disparate spatiotemporal dynamics. The proposed architecture attempts to overcome this challenge by piecing together relevant neural modules (e.g., the temporal, graph, and subtask layers) capable of jointly capturing complex dynamics. This method is compared to many competing baseline models on several common time series benchmarks, as well as on a newly proposed Multiphysics Building System (MBS) dataset.

**Strengths:**

- The paper's stated contributions are clear and address a difficult, high-impact problem in the domain of multi-physics systems. The approach potentially lays the groundwork for tackling broader challenges across connected physics models (not just the building environment).
- The presentation of the paper is clear and well-organized. I appreciated the comprehensive literature review and logically grouped discussions across Sections 1-4, which made problem setup and methodology easy to compartmentalize and digest.
- The reported evaluation is very extensive, covering a variety of important dimensions that help position the model's utility. For instance, the model is compared to several baseline methods on common time series datasets (highlighting its comparative advantages), key ablations are reported (justifying architectural decisions), and different model sizes are evaluated, highlighting the impact of parameter scaling.

**Weaknesses:**

- The empirical evaluation of the method on multi-physics settings is somewhat limited, provided only the multi-physics building setting is explored. While results appear strong and there are diverse dynamics present, it is difficult to assess the proposed architecture's general utility as a multi-physics model beyond this domain.
- The analysis of empirical results would benefit from a discussion that characterizes when a complex architecture like HGTFT is justified, compared to a less complex model, such as LSTMs. It would be very insightful to see practical tradeoff considerations, for instance, highlighting model differences in performance at fixed parameter counts or time spent training.
- It is claimed that the more common time series benchmark datasets don't capture the multi-domain complexity that HGTFT targets, but there is little to no explanation behind why the architecture underperforms other methods on these benchmarks (e.g., in Table 13). Presumably many of the mechanisms relevant for capturing complex multi-physics interactions would be beneficial in modeling complex multi-variate time series more broadly. Additionally, it is not particularly clear why the other datasets, e.g., traffic, are not considered to exhibit multi-scale dynamics among diverse entities (stated in Section 6.3) when these are common qualities of traffic forecasting settings. Characterizing the performance differences across these settings in more depth would go some way to helping bridge the empirical gap and help characterize model behavior in lieu of an additional multi-physics dataset.

**Questions:**

- In Table 2, there are a few counter-intuitive fluctuations in performance across zero-shot and few-shot settings. For instance, the RCS metric is better for HGTFT zero-shot than it is in the few-shot (in both the "50 MBS" and "Full MBS" settings). I understand minor fluctuations could very well be noise, but is there a more principled reason behind this?
- In the main multi-physics evaluation setup, building samples from the MBS dataset are used for pre-training. Is there a significant amount of overlap between MBS and BTS? What dynamics does MBS capture that are expected to be helpful for BTS, or perhaps more importantly, what are the meaningful differences (potentially highlighting ability to generalize to new dynamics seen only in BTS)?

---

> ### Author Response · Authors · 2025-11-20
>
> We appreciate the reviewer’s insightful feedback and provide the following responses to address the concerns raised.
>
> **Response to Weakness 1:**
>
> We acknowledge that the availability of large-scale, publicly accessible multi-physics datasets remains limited. To address this, our study focuses on the BTS and our newly released MBS datasets, which together integrate real and simulated data and capture a broad range of coupled physical processes.
>
> Importantly, modern buildings themselves constitute a multi-physics environment, encompassing hydronic systems, air-side HVAC, thermal envelopes, refrigeration loops, and control logic. Our model demonstrates consistent performance across these heterogeneous physical mechanisms. In Appendix J.4, we further provide sub-domain analyses over four canonical physical domains, and the zero-shot and few-shot results show that HGTFT generalizes well even when the governing physics differ substantially.
>
> Beyond the building domain, we also evaluate HGTFT on diverse spatiotemporal graph benchmarks—including PeMSD4/8 and COVID-19—demonstrating that the architecture’s graph–temporal fusion design remains broadly applicable even without using the RCS. When RCS loss is employed, additional gains are observed on datasets with richer relational heterogeneity.
>
> Collectively, these results indicate that HGTFT is not tailored to a single domain but provides a robust and generalizable framework for multi-physics and heterogeneous spatiotemporal modeling.
>
>
> **Response to Weakness 2:**
>
> In the revised paper, we added **Appendix J.5 (“Performance Gains Under Fixed Capacity”)** to analyze when a complex architecture like HGTFT is justified over simpler models such as LSTM. We consider two scenarios: (i) single-variable prediction on a univariate dataset (ETT) and (ii) multi-object prediction with two object types and multiple dynamic and static features. Results show that while HGTFT and LSTM perform similarly in the single-object setting, HGTFT clearly outperforms LSTM in the multi-object scenario, highlighting the benefits of graph–temporal fusion and heterogeneous interaction modeling. This demonstrates that model complexity is warranted when cross-entity interactions are significant.
>
> Additionally, Appendix D (Figure 8) provides a model scaling study showing the relationship between HGTFT size and forecasting performance. This study indicates that a 310M-parameter model achieves the best balance of performance and efficiency on the MBS dataset, supporting the choice of model capacity in our experiments.
>
>
> **Response to Weakness 3:**
>
> The HGTFT architecture, including the number of Transformer layers (three layers), was determined based on extensive experiments on the MBS dataset, balancing performance and computational efficiency (Appendix D). We found that a shallow configuration with three Transformer layers provides the best trade-off: increasing depth brings limited benefit due to the strong coupling between temporal and graph structures, which already enables rich spatiotemporal representation learning.
>
> For fairness and consistency, this configuration is applied uniformly across all datasets without per-task tuning. The datasets reported in Table 13 （Table 14 in the revised paper）are purely temporal benchmarks that do not contain graph structures or static attributes, and therefore cannot fully leverage HGTFT’s heterogeneous graph encoder and static–dynamic fusion modules. Consequently, while deeper architectures (e.g., six transformer layers) indeed improve accuracy on such purely temporal datasets like ETT—achieving results comparable to state-of-the-art baselines—this direction is not the primary focus of our study, which targets heterogeneous, graph-structured, and multiphysics forecasting.
>
> Notably, while graph-structured datasets like PeMSD4 and PeMSD8 include spatial and temporal dependencies, their spatial relations are largely homogeneous and stationary (e.g., fixed road networks and consistent traffic flow patterns). These datasets mainly reflect localized temporal fluctuations within a single physical domain, where interactions across nodes remain relatively stable and unidimensional. In contrast, the COVID-19 dataset exhibits genuinely multi-scale and cross-domain dynamics, driven by non-stationary interventions (e.g., lockdowns and policy shifts), heterogeneous regional attributes (e.g., population density, healthcare capacity), and multi-relational dependencies (e.g., mobility flows, policy synchronization, and geographic proximity). These properties introduce time-varying, hierarchical interactions across entities—closer to the multi-physics coupling seen in our target applications.

---

> > ### Author Response · Authors · 2025-11-20
> >
> > **Response to Question 1:**
> >
> > This observation is insightful and not mere noise. Pre-training incorporates RCS in the physics-informed loss (Section 5.3, Appendix F.4), embedding strong physical constraints (e.g., operational bounds). Few-shot fine-tuning uses MSE-only (project-specific accuracy focus), slightly trading off RCS (0.0025 & 0.0018 zero-shot vs. 0.0037 & 0.0032 few-shot on BTS) for MSE gains (0.0056 & 0.0047 to 0.0036 & 0.0033). This reflects the practical objective of fine-tuning—prioritizing predictive accuracy, even at the cost of a small reduction in physical consistency.
> >
> > Furthermore, it can be observed that without pretraining, the model exhibits much poorer physical consistency, as it has not learned from the physics-informed loss. The fine-tuned model with pretraining achieves lower MSE than the model trained from scratch, demonstrating that large-scale physics-aware pretraining enables the model to capture richer physical representations, which in turn improves its ability to find more accurate prediction outcomes.
> >
> > **Response to Question 2:**
> >
> > - Coverage: MBS includes all major object and relation types that appear in BTS (HVAC components, thermal zones, control signals), which enables zero-shot forecasting directly on BTS.
> >
> > - Meaningful differences: Despite this structural overlap, the two datasets differ substantially in non-physical factors: BTS buildings are located in Australia, whereas the real data behind MBS comes from China. As a result, control strategies, occupancy schedules, climate conditions, and building usage patterns differ significantly.
> >
> > - What MBS captures that is helpful: MBS provides generalizable multi-physics priors—thermal–fluid interactions, sensor–actuator dependencies, and cross-entity dynamics—that are project-agnostic and transferable to new buildings.
> >
> > - What differs and why generalization is non-trivial: BTS contains new temporal regimes (e.g., climate-driven loads, different operation schedules), so the ability of HGTFT to zero-shot on BTS indicates that the model is not memorizing dataset-specific patterns but learning structural physical relationships across entities.
> > (5) Generalization conclusion: These observations demonstrate that as long as the physical object types and their interactions are represented in the pre-training corpus (MBS), the model can effectively generalize to new operational profiles seen only in BTS.
> >
> > We hope that our responses and the additional experimental validation have adequately addressed the reviewer’s concerns.

---

### Official Review · Reviewer_iQBU · 2025-11-04

**Soundness:** 2
**Presentation:** 1
**Contribution:** 4
**Rating:** 2
**Confidence:** 4

**Summary:**

The paper proposes a multiheaded GAT to model heterogeneous spatiotemporal graph data. The authors also examine the effectiveness of masked pre-training. The training pipeline is validated on a simulated heterogeneous spatiotemporal dataset.

**Strengths:**

- Originality and Significance: The paper addresses the lack of heterogeneous graph datasets in the field of spatiotemporal forecasting.
- Clarity: The multiple technical contributions are well explained.
- Significance: The paper touches on masked pre-training and analyzes zero-shot and few-shot performances.

**Weaknesses:**

- Literature Review:
  - The paper does not provide physics-prioritized methods
  - The paper misses a few important related works. E.g.
    - DCRNN: Li, Y., Yu, R., Shahabi, C., & Liu, Y. (2017). Diffusion convolutional recurrent neural network: Data-driven traffic forecasting. arXiv preprint arXiv:1707.01926.
    - STEP: Shao, Z., Zhang, Z., Wang, F., & Xu, Y. (2022, August). Pre-training enhanced spatial-temporal graph neural network for multivariate time series forecasting. In Proceedings of the 28th ACM SIGKDD conference on knowledge discovery and data mining (pp. 1567-1577).
- Quality:
  - The paper has too many directions, making it hard to follow the central contribution. The authors might benefit from focusing on heterogeneous graphs only.
- Clarity:
  - The paper's wording is vague, making it hard to follow. E.g. in the following text, the authors could provide examples of different node types interacting simultaneously.
  > More generally, multi-domain physical systems such as power grids and building operations, where heterogeneous entities interact across multiple physical fields. Accurate forecasting in such systems is critical for efficiency, safety, and sustainability, yet remains challenging due to diverse data modalities, structural dependencies, and domain-specific physical mechanisms.
  - Some typos. e.g. in section 6.1,
  > Mult-domain physical System Datasets

**Questions:**

- How is LSTM pretrained and evaluated zero-shot in your framework?
- Could we use the output from the pre-trained models as input to other baseline methods? Would this approach increase the accuracy compared to using a linear projection from node values to the hidden space?

---

> ### Author Response · Authors · 2025-11-20
>
> Thank you for recognizing the value and contributions of our work. We appreciate the reviewer’s insightful feedback and provide the following responses to address the concerns raised.
>
> **Response to Weakness 1:**
>
> - We added a paragraph of literature review for "Physics-prioritized methods" on page 2 Line 106.
> - We also added DCRNN and STEP in the paragraph for "Spatial-Temporal Forecasting" on page 2 Line 85.
>
> Please see the revised paper.
>
> **Response to Weakness 2:**
>
> We appreciate the reviewer’s feedback. While our paper indeed addresses multiple aspects, we believe all contributions are closely interrelated and essential to the central problem of heterogeneous graph forecasting in multi-domain physical systems. In particular:
>
> - The focus on heterogeneous graphs defines the problem scope, as real-world infrastructures involve diverse entities and relationships.
> - HGTFT, with tailored tokenization and embedding, provides the algorithmic foundation to model these heterogeneous structures effectively.
> - The physics-informed training pipeline ensures that predictions are not only accurate but also consistent with the underlying physical mechanisms, which is critical for real-world systems.
>
> These components are designed to work together: the heterogeneous graph perspective motivates the modeling approach, the model design addresses graph complexity, and the training pipeline integrates domain knowledge to enhance reliability. Limiting the scope solely to heterogeneous graphs without these complementary innovations would omit the key novelty and practical significance of our work.
>
> **Response to Weakness 3:**
>
> In the revised version (Page 1 Line 32), we have rewritten the paragraph to include concrete examples and improve readability. The updated text now reads as follows:
>
> “More generally, multi-domain physical systems such as power grids and building energy networks consist of heterogeneous entities (e.g., sensors, actuators, and control devices) that interact across multiple physical fields. Accurate forecasting in these systems is critical for efficiency, safety, and sustainability, yet remains challenging due to diverse data modalities, complex structural dependencies, and domain-specific physical dynamics.”
>
> We have also revised several phrases, including the typo reviewer mentioned.
> Line 033: "power grids and building operations" → "power grids and building energy networks"
> Line 174: "The Temporal Layer depicts processing for a single object type, shared across all objects." (add "type" for precision).
> Line 327: "Multi-domain physical System Datasets." (correct the typo)

---

> > ### Author Response · Authors · 2025-11-20
> >
> > **Response to Question 1:**
> >
> > To provide a fair comparison with our HGTFT model, which benefits from large-scale pre-training, we adapt LSTM to a similar pre-training pipeline where applicable, treating it as a non-graph-aware model that processes each object type independently (i.e., without using static attributes or relational graph information, as noted in Appendix G). Specifically, we apply the masked time-series modeling pretext task and standard forecasting task using the previous 7 days (672 timesteps at 15-minute intervals) to predict the next day (96 timesteps). Each object type is modeled independently, using all its dynamic variables, with the input-output specification kept fully consistent with the HGTFT setup.
> >
> > In zero-shot settings, the pre-trained LSTM (from MBS) is directly applied to downstream datasets (e.g., BTS) without any further training or adaptation on the target dataset. Predictions are made independently for each object in the target data. This setup ensures a fair zero-shot comparison with HGTFT, isolating the effect of pre-training while highlighting the advantage of incorporating graph structures and static features.
> >
> > **Response to Question 2:**
> >
> > We appreciate the reviewer’s insightful question. We understand that this suggestion aims to disentangle the effects of representation quality from model architecture. Specifically, two complementary validations can address this:
> >
> > - **Decoder replacement:** using the embeddings produced by the pre-trained HGTFT encoder while substituting the downstream subtask network with alternative decoders to evaluate whether performance gains primarily stem from the embedding quality or decoder design.
> >
> > In ablation study (shown in Table 4), we have done two tests of replacing Decoder (Subtask:w/o GRU, Subtask: dense) simplifying decoder worsens MSE by 78–148%.
> >
> > We also added tests, frozing the pre-trained HGTFT encoder and replaced the subtask decoder with alternatives: a simple MLP (multi-layer perceptron, akin to our Subtask:dense ablation) and an LSTM decoder (2-layer bidirectional LSTM. The comparison has been conducted for the same data set for Table 3 and Table 4. The results are shown in the table below:
> >
> > | Metric |  HGTFT | Subtask: w/o GRU | Subtask: dense | Subtask: MLP | Subtask: LSTM |
> > |--------|--------|------------------|----------------|-------------|--------------|
> > |   MSE  | 0.0027 | 0.0048 | 0.0067 | 0.0034 | 0.0032 |
> > |   RCS  | 0.0012 | 0.0136 | 0.0297 | 0.0037 | 0.0031 |
> > |   CRS  | 0.3123 | 0.4622 | 0.5435 | 0.3586 | 0.3479 |
> > |   FDS  | 0.4052 | 0.4961 | 0.5803 | 0.4443 | 0.4203 |
> >
> > From the results, we can see that the HGTFT pretrained encoder with different replacements can have a good accuracy comparison with other baseline methods, but the proposed subtask network is slightly better.
> >
> >
> > - **Embedding comparison:** replacing the HGTFT graph-aware embeddings with simple linear projections from node features to the hidden space, while keeping the same subtask network, to measure the contribution of the learned embeddings.
> >
> > In ablation study, we have done several tests adjusting embeddings (Fusion:dense, Graph:removed, Temporal:removed), and the results are shown in Table 4.
> >
> > In addition, we further simplified the embeddings with linear projections from raw node values directly to the hidden space and keep the full subtask decoder fixed.
> > The results for Linear Projections + HGTFT Decoder: MSE=0.0058, RCS=0.0273, CRS=0.4912, FDS=0.4498
> > The result shows worse comparing to the ablation study results in Table 4, which proves necessary for each component in HGTFT.
> >
> > We hope that our clarifications and the further experimental evidence will positively update the reviewer’s assessment of our work.

---

### Author Response · Authors · 2025-12-02
**Final Remark**

We sincerely thank all reviewers for recognizing the significance and potential impact of our work. In response to each reviewer’s concerns, we have provided detailed clarifications and conducted additional analyses and experiments, some of which have been incorporated into the revised manuscript. Below, we summarize our responses to the major concerns that are common across reviewers.

**1. Heuristic Physics Loss vs. True Physical Understanding** (Bu3n Weakness2 & Q1, W4Fj Weakness2)

We explicitly clarify that RCS/CRS/FDS are soft, domain-informed regularizers rather than governing-equation-based physics solvers, and we do not claim true physical understanding. Their role is to bias learning toward physically plausible regimes while keeping MSE as the dominant objective. We further show that HGTFT generalizes without RCS, and that adding RCS yields a consistent 10–18% MSE improvement in few-shot settings.

Evidence: Clarification in **Section 5.3** & **Appendix F.4**, staged loss schedule in **Table 11**, and performance gains in **Table 2** and **Appendix J.4**.

**2. Generalizability Beyond the Building Domain** (uK4E Weakness1, W4Fj Q2, Bu3n Q1)

To address concerns about domain specificity, we decompose the building system into four canonical physical sub-domains (hydronic loops, HVAC air systems, thermal envelopes, and refrigeration/plant systems), and evaluate the performance on each of them, demonstrating consistently strong results across all four sub-domains.

Evidence: Sub-domain zero-/few-shot results in **Appendix J.4**.

**3. Validation on Real, Chaotic Data vs. Simulator Overfitting** (W4Fj Q3 & Q4)

Appendix A.1 serves only as a convergence sanity check using a simple FCU example, while Appendix A.2 demonstrates a more complex nonlinear multi-zone HVAC system representative of real-world dynamics. Our main evaluations rely on real-world datasets, including BTS and MBS with 1,045 real buildings. Large-scale simulation is used only to augment scenario diversity, not to replace real physics, and is physics-based and calibrated with real data.

Evidence: MBS dataset description in **Appendix B**, zero-/few-shot results on BTS in **Table 2**, and ablations in **Table 4**.

**4. Baseline Fairness and Training Protocol** (W4Fj Weakness1 & Q1, iQBU Q1)

We clarify that all channel-independent baselines follow exactly the same masked pretraining + forecasting pipeline as HGTFT, trained per object type using all dynamic variables with identical input–output specifications. Beyond the representative baselines reported in Tables 1–4, we additionally include a broad set of recent spatio-temporal forecasting baselines in Appendix G (Table 13) for further comparison.

Evidence: Training protocol clarification in **Appendix G**, and baseline comparisons in **Table 1-4**, and **Table 13**.

**5. Limited Scaling Gains and Model Complexity Justification** (W4Fj Weakness3、uK4E Weakness2)

The minimal gain from 310M → 1.26B is due to a predictability ceiling in complex physical systems, not poor architectural scalability. We further show that HGTFT’s advantage appears specifically in multi-object, multi-entity settings, whereas simpler models suffice in single-object cases.

Evidence: Scaling study in **Appendix D** (Figure 8 & Table 8) and complexity–benefit analysis under fixed capacity in **Appendix J.5**.

---

### Meta-Review · Area_Chair_zLZE · 2026-01-06

**Summary:**

The paper proposes HGTFT, a graph transformer architecture for time series forecasting in multi-physics domains. Through optimized normalization and physics-informed loss functions, HGTFT achieves strong performance in building environments.

The rebuttal convincingly addressed the majority of reviewers’ concerns. In particular, the authors conducted many experiments, such as a new variant of proposed method (Reviewer iQBU), the scenarios suitable for the proposed method (Reviewer uK4E), automatic loss-weight learning for losses, and comparison of normalization methods (Reviewer Bu3n).

However, the technical contribution remains incremental in nature, as the individual modules have all been proposed in prior work and the primary contribution lies in their integration. Moreover, the proposed method is specifically effective in multi-physics settings, which inherently limits its scope of applicability. As a result, only a small number of datasets are suitable for demonstrating its advantages. Indeed, as shown in Table 14, on standard time-series forecasting benchmarks such as weather, electricity, and traffic datasets, the proposed method underperforms compared to recent approaches.

Overall, while the paper presents a solid and well-executed study, it remains slightly below the acceptance bar.

**Reviewer Concerns:**

Reviewer iQBU raised the concerns about related works and baselines, presentation issues and typos, implementation details such as the pertaining of LSTM, and a new variant of proposed method.

Reviewer uK4E questioned about the limited scope of multi-physics settings, the comparison with LSTM under different scenarios, the fluctuations of RCS, and the transferability between MBS and BTS datasets.

Reviewer Bu3n expressed the concerns about the limited technical contribution, and heuristic physics-informed losses (RCS/CRS/FDS), out-of-distribution generalization, the weight among losses, and different normalization methods.

Reviewer W4Fj raised the concerns about the recent baselines, the use of evaluation metrics, the scalability of the proposed architecture, some implementation details, and the heavy reliance on simulated data.

The rebuttal convincingly addressed the majority of reviewers’ concerns. In particular, the authors conducted many experiments, such as a new variant of proposed method (Reviewer iQBU), the scenarios suitable for the proposed method (Reviewer uK4E), automatic loss-weight learning for losses, and comparison of normalization methods (Reviewer Bu3n).

However, the technical contribution remains incremental in nature, as the individual modules have all been proposed in prior work and the primary contribution lies in their integration. Moreover, the proposed method is specifically effective in multi-physics settings, which inherently limits its scope of applicability. As a result, only a small number of datasets are suitable for demonstrating its advantages. Indeed, as shown in Table 14, on standard time-series forecasting benchmarks such as weather, electricity, and traffic datasets, the proposed method underperforms compared to recent approaches.

**Reviewer Scores:**

Reviewer iQBU is likely to improve the score because most of concerns are addressed, while other reviewers may keep the scores.

---

### Decision · Program_Chairs · 2026-01-26

Reject